# Real-time visualization of clustering and intracellular transport of gold nanoparticles by correlative imaging

Mengmeng Liu[1,2,*], Qian Li[1,*], Le Liang[1], Jiang Li[1], Kun Wang[1], Jiajun Li[1], Min Lv[1], Nan Chen[1], Haiyun Song[3], Joon Lee[4], Jiye Shi[5,6], Lihua Wang[1], Ratnesh Lal[4] & Chunhai Fan[1]

Mechanistic understanding of the endocytosis and intracellular trafficking of nanoparticles is essential for designing smart theranostic carriers. Physico-chemical properties, including size, clustering and surface chemistry of nanoparticles regulate their cellular uptake and transport. Significantly, even single nanoparticles could cluster intracellularly, yet their clustering state and subsequent trafficking are not well understood. Here, we used DNA-decorated gold (fPlas-gold) nanoparticles as a dually emissive fluorescent and plasmonic probe to examine their clustering states and intracellular transport. Evidence from correlative fluorescence and plasmonic imaging shows that endocytosis of fPlas-gold follows multiple pathways. In the early stages of endocytosis, fPlas-gold nanoparticles appear mostly as single particles and they cluster during the vesicular transport and maturation. The speed of encapsulated fPlas-gold transport was critically dependent on the size of clusters but not on the types of organelle such as endosomes and lysosomes. Our results provide key strategies for engineering theranostic nanocarriers for efficient health management.

[1] Division of Physical Biology & Bioimaging Center, Shanghai Synchrotron Radiation Facility, CAS Key Laboratory of Interfacial Physics and Technology, Shanghai Institute of Applied Physics, Chinese Academy of Sciences, Shanghai 201800, China. [2] The Institute for Translational Nanomedicine, Shanghai East Hospital, Shanghai 200120, China. [3] Key Laboratory of Food Safety Research, Institute for Nutritional Sciences, Shanghai Institutes for Biological Sciences, Chinese Academy of Sciences, Shanghai 200031, China. [4] Department of Bioengineering, Materials Science and Engineering and Department of Mechanical and Aerospace Engineering, Institute of Engineering in Medicine, University of California San Diego, La Jolla, California 92093, USA. [5] Kellogg College, University of Oxford, Oxford, OX2 6PN, UK. [6] UCB Pharma, Slough, Berkshire SL1 3WE, UK. * These authors contributed equally to this work. Correspondence and requests for materials should be addressed to R.L. (email: rlal@ucsd.edu) or to C.F. (email: fchh@sinap.ac.cn).

Endocytosis and vesicle transport are vital cellular processes in all eukaryotes[1,2]. Typically, cellular internalization initiates a series of events including the formation of early endosomes, their sorting and maturation, and subsequent directed transport[3,4]. Studies of these processes provide not only knowledge on how cells communicate with the outside world and receive nutrients and signals, but also insights about the invasion mechanism of viruses or artificial micro-/nano-agents[5–10]. Because of their significance in both fundamental cell biology and potential applications in pathology (for example, neurodegenerative or infectious diseases)[11] and theranostics (for example, drug delivery)[12,13], extensive theoretical and experimental studies have been carried out[14–17]. The importance of nanoparticle characteristics such as size, shape and surface chemistry in their cellular interactions have been reported[18]. Yet, the mechanisms and roles of nanoparticle clustering in endocytosis and intracellular traffic remains largely unexplored[19–21]. Given that clustering of nanoparticles is a widespread phenomenon in solution and especially in cellular environments, it is highly significant to understand real-time clustering of nanoparticles and how clustered nanoparticles are transported inside a cell.

Imaging-based techniques, primarily electron microscopy (EM) and fluorescence microscopy (FM), provide simple and powerful approaches to examine intracellular trafficking pathways[1,22–24] though their role in real-time imaging is limited. FM, especially with the advent of total internal reflection fluorescence microscopy and epi-FM, allow direct, non-invasive monitoring of dynamic motions of endocytosed vesicles in real time[15]. Despite its unparalleled advantages in cellular imaging, FM heavily relies on efficient labelling of fluorophores, and has limited spatial resolution ($\sim$200–300 nm). Recent advances in super-resolution fluorescence microscopy (SR-FM), including photoactivated localization microscopy, stochastic optical reconstruction microscopy and stimulated emission depletion, have improved resolution remarkably and opened new opportunities for studying spatiotemporal features of subcellular vesicles[25–27]. Nevertheless, SR-FM also has several restrictions, including low temporal resolution, the use of high-power lasers or limited choice of fluorophores[28,29]. In addition, intensity-based FM imaging is intrinsically susceptible to environmental changes that restricts its application in studying clustering of nanoparticles and resolving relative movements of multiple vesicles[21].

Metallic nanoprobe-based plasmonic imaging has emerged as an alternative to fluorescence imaging[30–34]. Metallic nanostructures are intrinsically brighter than fluorophores and essentially free of blinking and photobleaching[35]. They provide subwavelength localization of surface plasmon-polaritons (SPPs), which in turn provides an unique opportunity for imaging sub-diffraction limited structures[36]. Significant spectral wavelength shift in plasmon coupling of nanoparticles[37] would allow monitoring the clustering state of nanoparticles in cells. Since gold nanoparticles (AuNPs) are ready to be endocytosed with high efficiency[38–40], they serve as intrinsic and non-bleachable probes for long-term imaging of vesicular transport at single-particle level without prior labelling of nanoparticles. In this study, we designed DNA-decorated gold (fPlas-gold) nanoparticles as a dually emissive fluorescent and plasmonic probe and examined their cellular uptake and intracellular trafficking using a correlative FM and plasmonics-based dark-field microscopy (DFM). This study reports the first direct relationship between clustering states of nanoparticles and their real-time intracellular transport at the single-cell level (Supplementary Fig. 1). The fPlas-gold nanoprobes appeared mostly as single particles in the early stage of endocytosis and then clustered during the transport and maturation process in cells; their clustering remarkably affected their velocity of intracellular transport but was independent of the types of cellular organelles.

## Results

**Synthesis and characterization of fPlas-gold**. A fPlas-gold nanoparticle contains a gold core ($\sim$50 nm AuNP) for SPP guidance and a loose DNA shell tagged with a fluorophore (Cy3). To fabricate this nanoprobe, first thiolated DNA strands were self-assembled to AuNPs and then hybridized with Cy3-tagged complementary strands (Supplementary Fig. 2). Compared to bare AuNPs, Z-potential of fPlas-gold increased from $-44.0 \pm 0.4$ to $-34.4 \pm 0.4$ mV, and the diameter of fPlas-gold increased from $46.3 \pm 0.5$ to $71.6 \pm 1.8$ nm. The increase in the hydrodynamic diameter of AuNPs from the dynamic light scattering measurements corresponds to an adlayer of DNA. In addition, the spacing layer of rigid dsDNA minimized metal-induced fluorescence quenching, resulting in a fluorescent nanoprobe emitting red fluorescence. Also, the modification of AuNPs with dsDNA resulted in a slight redshift of 2 nm in Ultraviolet–vis absorption; whereas no aggregation peak was observed, suggesting high water-dispersity of the nanoprobe (Supplementary Fig. 3a). Transmission electron microscopy (TEM) studies further confirmed that this nanoprobe retained its monodispersity (Supplementary Fig. 3b).

Correlative FM/DFM was used to image fPlas-gold nanoparticles anchored on glass slides or trapped in cells. Individual fPlas-gold nanoparticles displayed clear green colour in DFM (Supplementary Figs 3c,d and 4a). Comparison of the FM/DFM images taken for the same field of view indicates that the scattering intensity of fPlas-gold was much higher than fluorescent emission of Cy3. Further analysis revealed that several dots that were present in DFM images were missing in FM images, which might arise from the photobleaching of Cy3 or insufficient sensitivity of FM. Similar phenomenon was observed for intracellular fPlas-gold nanoparticles as well (Supplementary Fig. 4b). Hence, whereas FM remains to be a convenient and powerful method, DFM provides a complementary means to image nanoparticles with better sensitivity.

Given that the plasmonic resonance of AuNPs is critically dependent on their clustering state[35–38], the clustering effects of fPlas-gold nanoparticles was examined by DFM. First, closely packed fPlas-gold clusters were prepared and their size, shape and identity were confirmed by scanning electron microscopy (SEM) imaging. Then, *ex-situ* DFM image of individual clusters was obtained (Supplementary Fig. 6). Significantly, a single particle of fPlas-gold exhibited green colour, whereas the colour of clustered fPlas-gold gradually turned to yellow along with the increased number ($n$) of particles (from $n = 1$ to $n = 10$) (Supplementary Fig. 7a). Scattering spectra show that the peak wavelength gradually shifted to the red from 570 to 620 nm (Supplementary Fig. 7b). The spectra calculated from finite-difference time-domain simulation were generally consistent with the measured ones, except that the experimental spectra exhibited broadened peaks (Supplementary Fig. 7c). Hence, it is reasonable to ascertain that different size of fPlas-gold clusters resulted characteristic colours and scattering wavelengths. This property was used for visualizing aggregation states of fPlas-gold in cellular environment. Typically, it was easy to distinguish single particles ($n = 1$; green), small clusters ($n = 2–5$; yellow) and large clusters ($n > 5$; bright yellow) under DFM; these particles exhibited characteristic scattering spectra with maximum scattering wavelengths at $\sim$560, $\sim$580 and $\sim$610 nm, respectively (Supplementary Fig. 8). Furthermore, the scattering spectra of the

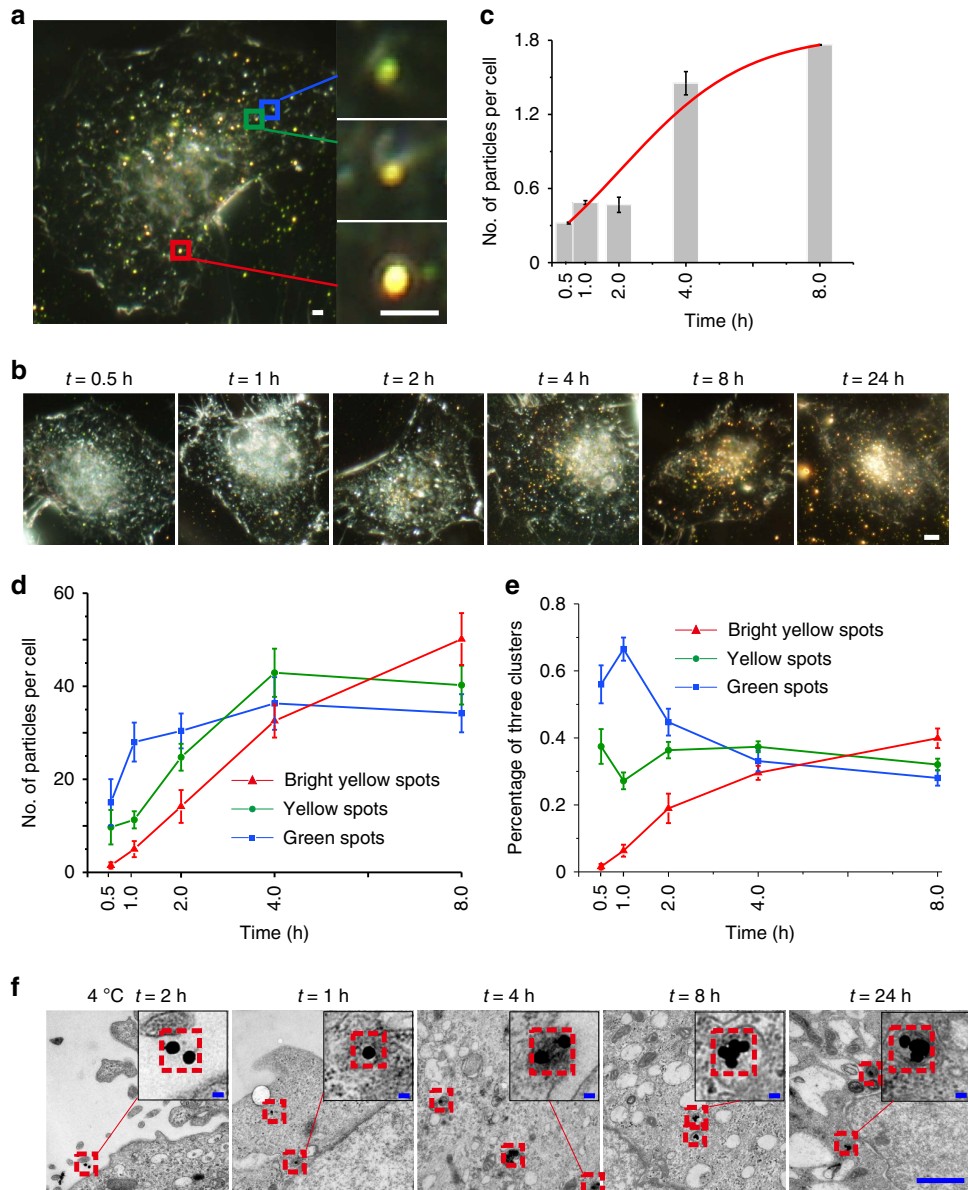

**Figure 1 | Cell entry of fPlas-gold.** (**a**) A representative dark-field microscopy (DFM) image showing the presence of single particles (green spots in blue rectangle), small clusters (yellow spots in green rectangle) and large clusters (bright yellow spots in red rectangle), respectively. Scale bar, 2 μm. (**b**) DFM and (**f**) transmission electron microscopy (TEM) images for time evolution of fPlas-gold incubated with HeLa cells. Scale bar in **b**, 5 μm; scale bar in **f**, 2 μm, inserted 50 nm. (**c**) Inductively coupled plasma atomic emission spectrometry (ICP-AES) analysis of elemental gold in cells along with the time (t). (**d**) Averaged particle counts and (**e**) percentages of different clustering states of fPlas-gold in cells over time (t). Data were analysed from 20 cells.

extracellular single particle matched well with the intracellular one (Supplementary Fig. 9), suggesting that the cellular microenvironment does not significantly affect the optical scattering properties (that is, colour) of fPlas-gold.

**Internalization and clustering of fPlas-gold in cells.** Having established the plasmonic properties of fPlas-gold, their internalization in living cells was examined. HeLa cells were incubated with fPlas-gold for 2 h and then internalized fPlas-gold nanoparticles were analysed by several methods. Both FM and DFM data showed that fPlas-gold could be rapidly and efficiently taken up by cells in a time-dependent manner (Fig. 1a,b and Supplementary Fig. 10). The amount of elemental Au in cells was evaluated by inductively coupled plasma atomic emission

spectrometry (ICP-AES) and showed the time-dependent accumulation of Au in cells (Fig. 1c). DFM imaging shows three types of dots with different colours. Interestingly, green dots representing single particles were mainly present surrounding the cellular membrane after 2 h incubation; yellow dots for small clusters randomly distributed in the cytoplasm; whereas bright yellow dots for large clusters were mostly localized to the perinuclear region. Significantly, there was no presence of Cy3 fluorescence signal detached from AuNPs or apparent separation of Cy3 fluorescence and AuNPs signal, implying that the observed florescence signals were from the assembled fPlas-gold (Supplementary Fig. 10).

To better understand the internalization and clustering of fPlas-gold, the clustering states of fPlas-gold were analysed outside and inside of the cell in 20 randomly selected cells

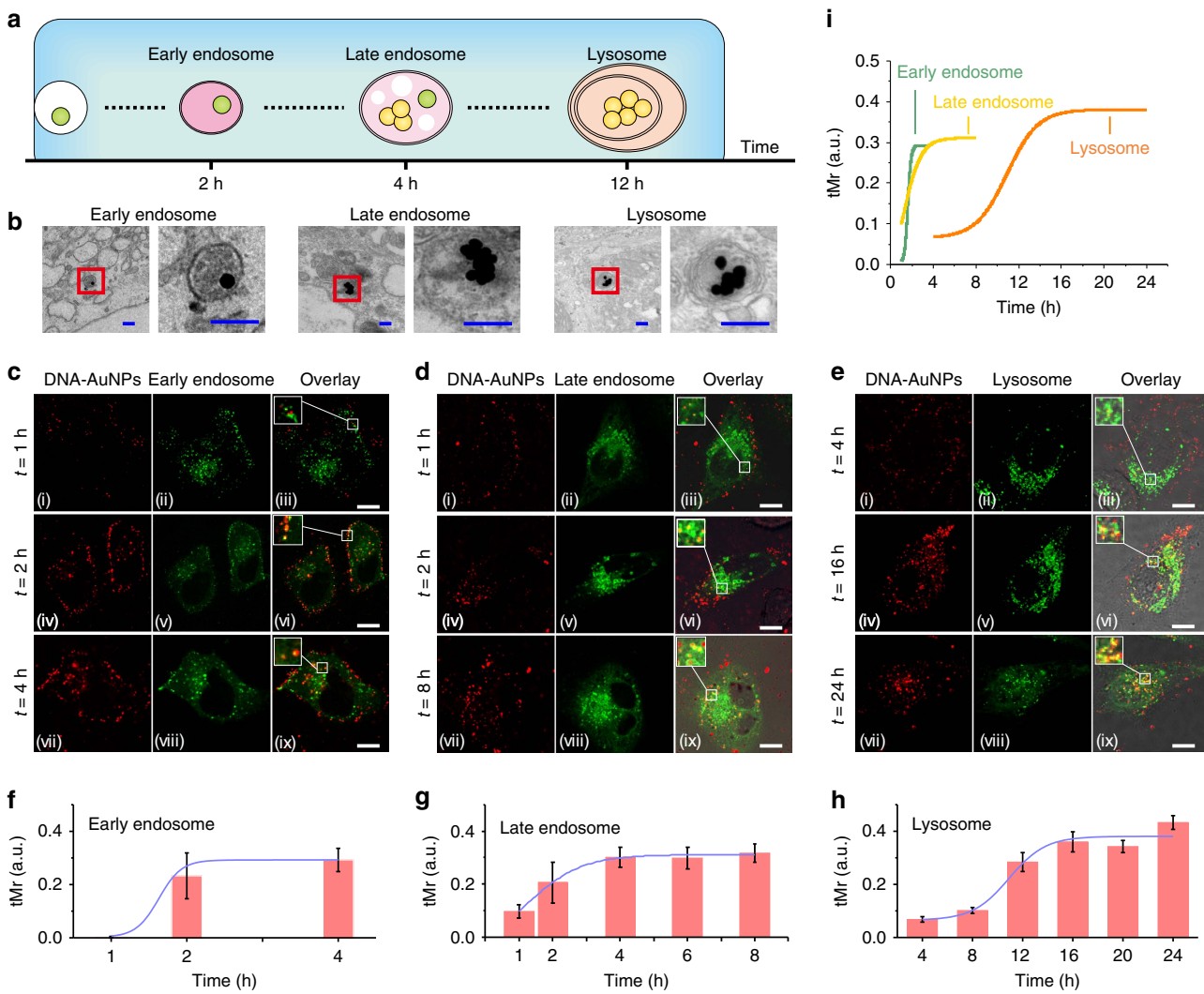

**Figure 2 | Intracellular trafficking of fPlas-gold.** (**a**) Schematic showing of intracellular trafficking of fPlas-gold. (**b**) Representative TEM images of fPlas-gold trapped in early, late endosomes and lysosomes. Scale bar, 200 nm. (**c–e**) Fluorescence microscopy (FM) images showing co-localization of fPlas-gold (red) with early endosomes (green), late endosomes (green) and lysosomes (green). Scale bar, 10 μm. (**f–i**) Plots of thresholded Mander's co-localization coefficient (tMr) values over time for early, late endosomes and lysosomes, respectively.

(Supplementary Fig. 11). The number of all dots increased along with the incubation time (Fig. 1d,e), indicating the time-dependent cellular uptake. However, the number of clusters increased much faster than that of single particles: ~60% single particles and ~40% small clusters at 0.5 h and ~70% small or large clusters at 8 h. These data suggest that clustering of fPlas-gold occur during the intracellular trafficking. Interestingly, over 90% of the extracellular dots were of green colour after 2 h incubation; whereas ~50% of intracellular dots were in the form of clusters (yellow or bright yellow, Supplementary Fig. 11), suggesting that fPlas-gold existed predominantly in the non-aggregated single particle state outside of the cell. TEM results revealed that fPlas-gold nanoparticles were mostly found in endosomal vesicles (Fig. 1f), suggesting that the formation of clusters possibly results from endocytosis and intracellular vesicle transport rather than self-aggregation of the nanoparticles.

**Internalization mechanism and intracellular traffic of fPlas-gold.** The internalization mechanisms of fPlas-gold were examined. A profound reduction in cellular uptake was observed when cells were incubated at 4 °C (Supplementary

Fig. 12), suggesting that the internalization is energy dependent. Moreover, endocytosis of fPlas-gold does not appear to follow any single, specific pathway, rather use multiple pathways including caveole- and clathrin-mediated endocytosis as well as macro-pinocytosis[41], although the clathrin-mediated endocytosis was the dominant one (Supplementary Figs 12 and 13).

Co-localization techniques were used to investigate the intracellular trafficking of the endocytosed nanoparticles. Red fluorescence of Cy3 and plamonic signal in fPlas-gold with green fluorescence of green fluorescent protein (GFP) fused to two specific protein markers of endosomes, Ras-related proteins 5 &7 (Rab 5 and Rab 7)[12] were observed, suggesting the formation of early and late endosomes, respectively (Fig. 2c,d and Supplementary Figs 14,15). The co-localization of fPlas-gold with early endosomes was found to be near the cellular membrane, and of late endosomes surrounded the nuclei. Also the co-localization of fPlas-gold with lysosomes that were stained with lysotracker, a selective dye for lysosomes[1], occurred mainly around the perinuclear region, suggesting that fPlas-gold nanoparticles were eventually transported to lysosomes (Fig. 2e). Consistent with these co-localization studies, TEM images (Fig. 2b and Supplementary Fig. 16) showed the corresponding fPlas-gold-

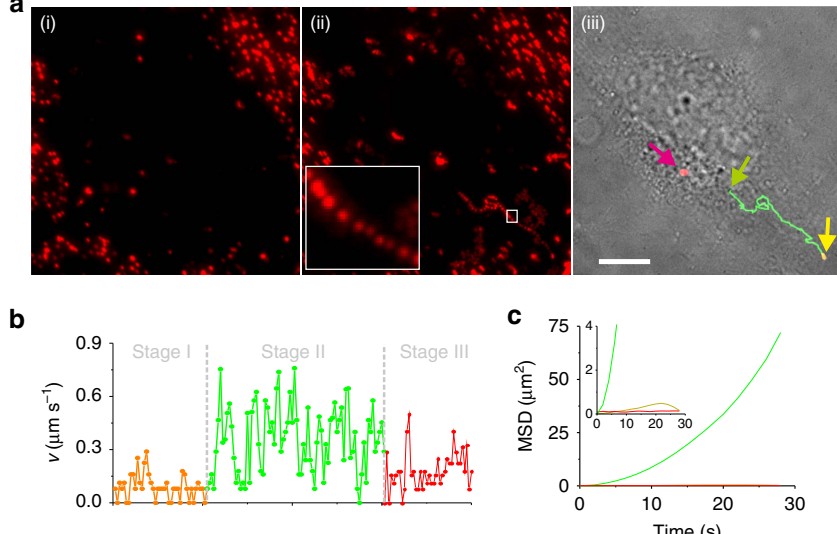

**Figure 3 | Long-term tracking of the transport of fPlas-gold.** (**a**) Uptake of fPlas-gold by a HeLa cell as visualized with a total internal reflection fluorescence (TIRF) microscope. (**a**(i)) Average time projection (Avg. Proj.) and (**a**(ii)) maximum time projection (Max. Proj.) collected from 120 frames of time-lapse microscopy (one frame per 2 s, with a total of 240 s). Inset shows the magnified image of the boxed region. (**a**(iii)) Bright field image for the trajectory of fPlas-gold. Scale bar, 10 μm. (**b**) The intracellular transport of fPlas-gold is displayed in three stages: (I) Low-motility particles representing those adhered to the membrane or bound to receptors (yellow); (II) High-motility particles representing those particles wrapped in early endosomal vesicles (green); (III) Low-motility particles wrapped in late endosomes or lysosomes in the perinuclear region (red). (**c**) 2D mean square displacement (MSD) analysis of the movement of fPlas-gold in the three stages shown in **b**. The super-linear trajectory curve (green) indicates microtubule-dependent movement of high-motility particles.

containing vesicles characteristic of early endosomes (~200 nm in diameter), late endosomes (~500 nm in diameter) and lysosomes (multi-layer spherical vesicles with up to μm-sized diameter). Significantly, fPlas-gold largely existed in the form of single particles in early endosomes, and they gradually became clustered during the transportation to the late endosomes and lysosomes (Supplementary Fig. 16). Time-lapse studies revealed the degree of co-localization of fPlas-gold with these cellular compartments, which were quantified by the thresholded Mander's co-localization coefficients (tMr)[42] representing the percentage of the Cy3 fluorescence overlaid with the GFP. The amount of trapped fPlas-gold reached the maximum level in early endosomes, late endosomes and lysosomes at 2, 4 and 16 h, respectively (Fig. 2f–i). Such a time-evolution reflects the sequence of intracellular traffic of fPlas-gold from early endosome to late endosome and eventually in lysosome.

**Dynamics of intracellular transport of fPlas-gold.** Having substantiated the intracellular traffic pathway of fPlas-gold, we examined the dynamics of fPlas-gold transport in live cells using FM imaging and two-dimensional single-particle tracking (Supplementary Movie 1: $\Delta t = 2$ s, total time $= 240$ s). By generating an average time projection of acquired time-lapse movies, which averages the same pixels in all the frames of the complete movie sequence to monitor static or slowly moving nanoparticles, the majority of fPlas-gold nanoparticles were confined in a small area (Fig. 3a(i)). The visualization of the moving nanoparticles show that some particles moving rapidly towards the perinuclear region (Fig. 3a(ii)). On the basis of these observations, a three-stage transport process for internalized fPlas-gold (Fig. 3a(iii),b) is proposed, that is, stage I: low motility (corresponding to fPlas-gold adhered to the membrane or bound to receptors); stage II: high motility (corresponding to fPlas-gold wrapped in early endosomal vesicles); stage III: low motility (corresponding to fPlas-gold wrapped in late endosomes or

lysosomes in the perinuclear region). Then the transport dynamics of fPlas-gold and the movement of these three organelles were examined (Supplementary Fig. 17 and Supplementary Movies 2–4: $\Delta t = 3$ s, total time $= 90$ s for 2 and 3, 450 s for 4). Results consistently showed that most particles captured in early endosomes were of high mobility, whereas those in late endosomes and lysosomes were of low mobility.

Mean square displacement (MSD) analysis was used to analyse the tracking data (Fig. 3c). In the plot of MSD versus time ($t$), the downwards curves displayed in orange and red corresponded to the two low-motility stages (Fig. 3c), suggesting that fPlas-gold was in anomalous, undirected motions[43]. In contrast, the high-motility fPlas-gold showed a super-linear curve (displayed in green colour in Fig. 3), suggesting directed motions that are probably associated with microtubule-dependent transport[43]. Indeed, the co-localization of fPlas-gold with the GFP-stained microtubulin was observed in FM imaging (Fig. 4a). Pharmacological inhibitors were then used to confirm the dependence of microtubules. The treatment of cells with cytochalasin B that depolymerizes actins[41] did not affect intracellular mobility of fPlas-gold (Fig. 4b,c and Supplementary Movies 5 and 6); whereas the presence of nocodazole that depolymerizes microtubules[1,6] drastically reduced the mobility (Fig. 4d and Supplementary Movie 7). Analysis of the movement of fPlas-gold along microtubules (Fig. 4e) showed that the average speed was of 0.17 μm s$^{-1}$ (Fig. 5d and Supplementary Movie 8: $\Delta t = 3$ s, total time $= 300$ s); whereas the average speed of low-motility fPlas-gold was of 0.04 μm s$^{-1}$.

Since FM does not support the identification of clustering states of fPlas-gold, DFM was used to investigate the intracellular movement of differentially clustered fPlas-gold (Supplementary Movies 9 and 10: $\Delta t = 1$ s, total time $= 300$ s). In total, 100 green, 100 yellow and 100 bright yellow dots were randomly selected in three independent experiments. Results indicate that the movement of fPlas-gold is critically dependent on their clustering states (Fig. 5a–c and Supplementary Figs 18 and 19), which

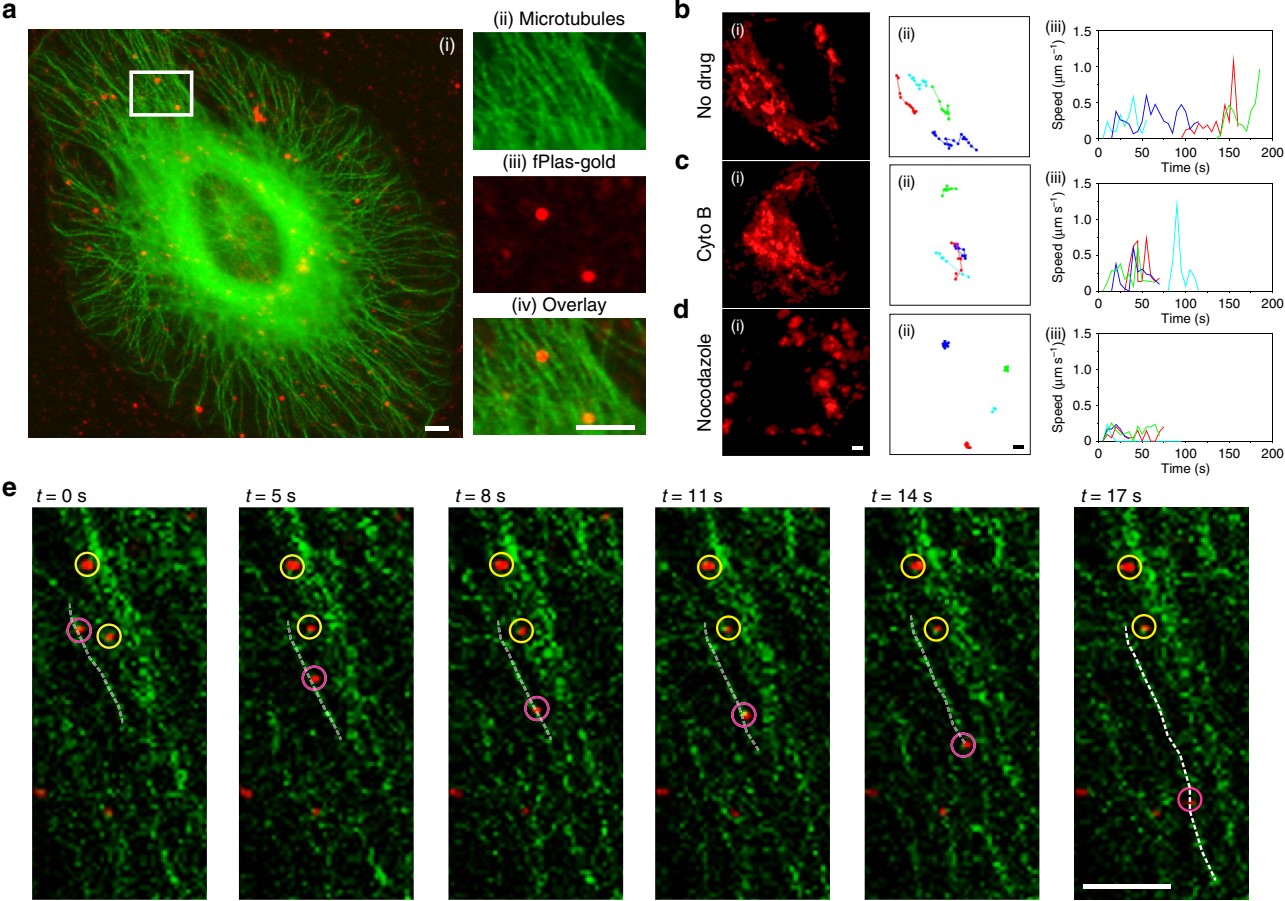

**Figure 4 | Microtubule-dependent transport of fPlas-gold.** (**a**) FM images of fPlas-gold (red) co-localized with GFP-stained microtubulin (green). Figures (**a**(ii)), (**a**(iii)) and (**a**(iv)) are enlarged images for the area marked with white rectangle in (**a**(i)). Cells with GFP-stained microtubulin were incubated without treatment (**b**), with 60 mM nocodazole (**c**) and 10 mM cytochalasin (**d**) for 30 min, respectively. Forty frames (one frame per 5 s, reflecting 200 s) of time-lapse microscopic images are displayed as maximal intensity projection (**b**(i), **c**(i) and **d**(i)). Representative trajectories are shown in (ii); and their instantaneous speeds as a function of time are shown in (iii). (**e**) Snapshots of fPlas-gold moving on the microtubule captured in FM, with 2D trajectory of a high-motility (rose pink circle) and two low-motility (yellow circle) particles. Scale bar, 4 μm.

falls into three categories. Interestingly, although two types of populations are observed in FM imaging, the average speed of movement of single particles observed under DFM was comparable to the high-motility population observed in fluorescence-based single-particle tracking (Fig. 5d), suggesting that most of single fPlas-gold nanoparticles were transported along the microtubule. MSD analysis also showed that the movement of large clusters was diffusion-like (Supplementary Fig. 19a), which, combined with TEM studies (Supplementary Figs 16 and 20), suggested that they might be formed when fPlas-gold was transported to the lysosome (Supplementary Fig. 16).

Statistical analysis of single-particle movement showed that both the passage length and the maximum frame to frame instantaneous speed decreased when single fPlas-gold gradually formed small clusters and large clusters (Fig. 5a–c). Movement of early endosomes, late endosomes and lysosomes was analysed by fluorescence-based single-particle tracking (Supplementary Figs 21–23). Of note, single fPlas-gold and the large clusters moved significantly faster than early endosomes and lysosomes, respectively, whereas the difference in speed between small clusters and late endosomes was insignificant. The passage length distribution of fPlas-gold was also different with that of vesicles (Supplementary Figs 21–23). Hence, comparison of the movement between fPlas-gold and organelles did not show a direct

correlation. Furthermore, contrary to the profound difference between movement of single fPlas-gold and small clusters (Fig. 5a–c), the movements of early and late endosomes are comparable (Supplementary Fig. 24). Therefore, although the motility of fPlas-gold might have some relationship with the movement of vesicles, it is primarily dependent on their clustering states.

The motions of high-motility fPlas-gold along the microtubule were monitored using DFM. By exploiting the plasmonic coupling effect, a reliable monitoring of the merging of single particles with the green-to-yellow colour change or small clusters with intensified scattering was possible. Interestingly, the data from these experiments show the presence of at least four different types of motions after 1 h incubation—that is, (I) chase and merge; (II) kiss and run; (III) back and forth; and (IV) stop shot. In motion I, a fast-moving single particle chased a slow-moving one (Fig. 6a, Supplementary Fig. 25 and Supplementary Movie 11). When the single fPlas-gold met each other, the colour changed from green to yellow and stayed intact over 4 min, suggesting that they merged into a small cluster. Since the plasmonic coupling occurs within ∼50 nm, this merging process probably arises from vesicular fusion of early endosomes. In motion II, when a single fPlas-gold (green) met a small cluster (yellow) on a different track, they transiently merged to a slightly brighter dot (yellow). Nevertheless, they rapidly separated to one

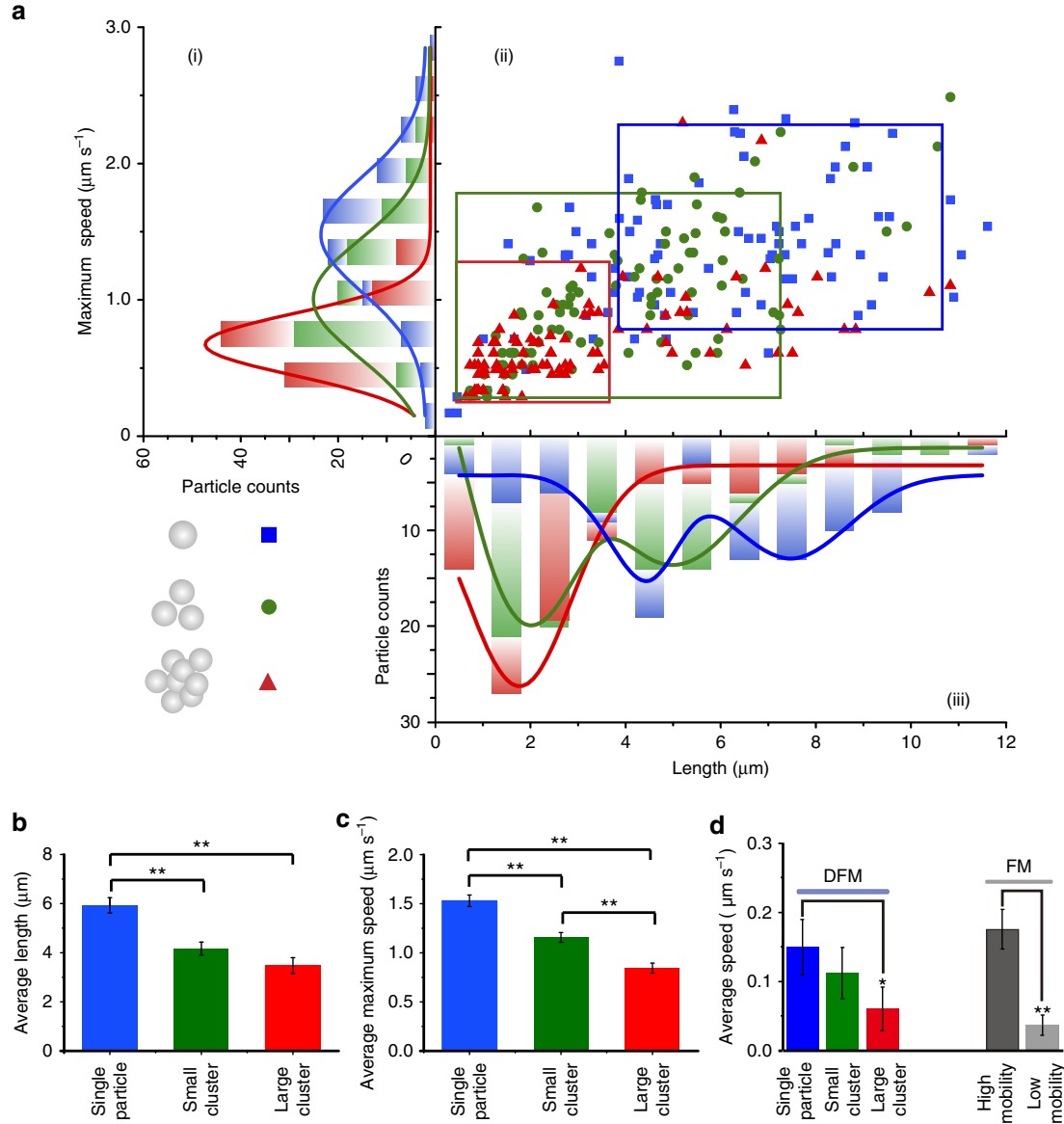

**Figure 5 | Movement of fPlas-gold in cells.** (**a**(ii)) Scatter plots showing passage length and maximum of the frame to frame instantaneous speed for each single particle (blue), small cluster (green) and large cluster (red). The rectangles represent the major distribution of these particles, containing 79 single particles (blue), 92 small clusters (green) and 75 large clusters (red), respectively. (**a**(i)) Histogram showing the distribution of maximum speed of single particles (blue), small clusters (green) and large clusters (red). (**a**(iii)) Histogram showing the distribution of passage length of single particles (blue), small clusters (green) and large clusters (red). Data were collected from 100 randomly selected spots in three independent experiments. (**b**) Average passage length and (**c**) average maximum speed of fPlas-gold (data obtained from single-particle analysis shown in **a**, and were presented as the mean ± s.e.m.). (**d**) Average speed ($v$) of fPlas-gold in cells. Particles observed using DFM were classified as single particles, small clusters and large clusters while the ones observed using FM were classified as high- and low-mobility ones, respectively. Corresponding curves of MSD are shown in Supplementary Fig. 19. Data were collected from 50 spots in three independent experiments for each group and presented as the mean ± s.e.m. *$P < 0.05$, **$P < 0.01$, according to two-tailed two-sample $t$-test.

green and one yellow dots again and continued to move along their original tracks (Fig. 6b, Supplementary Fig. 26 and Supplementary Movie 12). In motion III, a small cluster moved rapidly towards a static small cluster, which suddenly reversed its moving direction (Fig. 6c, Supplementary Fig. 27 and Supplementary Movie 13). This motion is associated with the well-known 'tug-of-war' bidirectional cargo transport along the microtubule driven by dynein and kinesin motors[16,43]. In motion IV, when two small clusters along two tracks in the cross-section of the microtubules, the fast-moving one lost its mobility and stayed static while the latter went away with high speed (Fig. 6d, Supplementary Fig. 28 and Supplementary

Movie 14). These different types of motions reflect the versatility of microtubule-dependent transport in cells, as well as the role of vesicle fusion in particle clustering (for example, motion I).

## Discussion

Results described in this study show that fPlas-gold nanoparticles enter cells mainly in the form of single particles. During the intracellular trafficking of fPlas-gold along microtubules to the perinuclear region, individual nanoparticles present in early endosomes were gradually clustered via vesicle fusion during the maturation process. When these nanoparticles enter lysosomes,

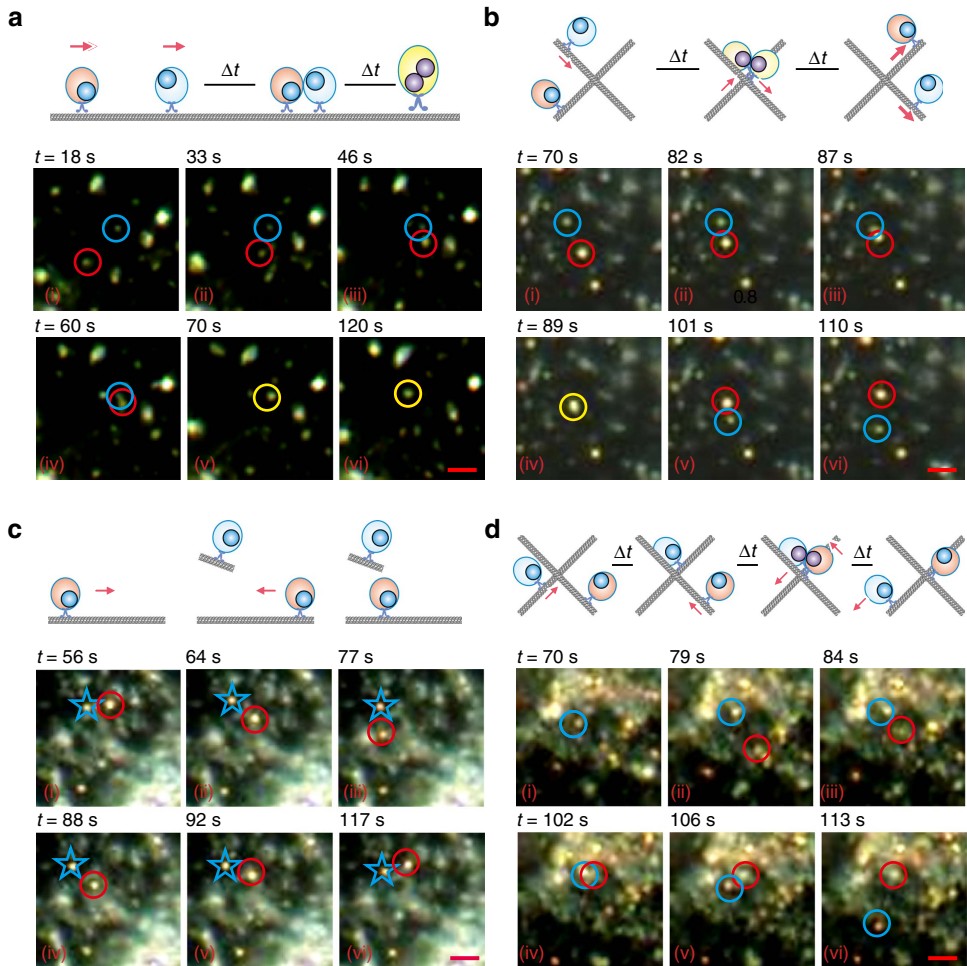

**Figure 6 | Motions of fPlas-gold along the microtubule as visualized with DFM.** (**a**) Chase and merge: one spot moved towards the other and fused into one spot. (**b**) Kiss and run: two spots moved head-on-head, and departed to the reverse directions upon collision. (**c**) Back and forth: one spot moved forwards and backwards to a static spot. (**d**) Stop shot: one spot moved towards the other, and stopped upon collision; while the other spot started moving to a different direction. Scale bar, 2 μm.

they exist predominantly as large clusters (Supplementary Fig. 1). Clustering of nanoparticles could affect their biological effects, including their interactions with cell membranes, receptor binding and targeting[20,21]; the results described here provide new perspectives for designing novel AuNP-based high-efficiency theranostic materials. Furthermore, 50 nm fPlas-gold appears to endocytosed via multiple energy-dependent pathways, although the clathrin-mediated endocytosis played a major role. Of note, these vesicles have various sizes, while fPlas-gold largely exists in the form of single particles in the vesicles, suggesting that the involvement of different pathways of cell entry is an intrinsic property of fPlas-gold that is independent of their own sizes. The dependence on multiple mechanisms for cell entry of AuNPs is often observed in viral infection, which might account for their high cellular uptake efficiency observed in previous studies[37–40].

The scattering signal obtained from DFM which reflects the plasmonic resonance coupling among AuNPs allowed the measurement of the distance within tens of nanometres. Hence, the clustering states of fPlas-gold nanoparticles in single cell could be identified. In FM studies, there are two populations of fPlas-gold, high motility and low motility, which are associated with the sorting process of internalized cargos.

Motility of fPlas-gold nanoparticles is appears to be critically dependent on their clustering states. Single fPlas-gold that moves rapidly falls within the high-motility population observed in FM

and corresponds to single particle-trapped early endosomes moving along the microtubule. Slow-moving large clusters fall within the low-motility population, corresponding to the large cluster-trapped late endosomes or lysosomes localized in the perinuclear region. Small clusters probably stay in the intermediate states during the maturation from early-to-late endosomes. When the movement between particles and organelles were compared, no direct correlation was found, suggesting that the organelle type does not significantly influence the nanoparticle movement. Notably, TEM analysis revealed that the size of vesicles containing individual nanoparticles was significantly smaller than that of small and large clusters (Supplementary Fig. 20), suggesting that the number of fPlas-gold in these vesicles corresponds well with the vesicle size.

Previous studies on virus/nanoparticle entry and movement in live cells generally fall into three categories based on imaging techniques employed, that is, fluorescence imaging, plasmonic imaging and correlative imaging. A series of earlier work on entry mechanisms and movement of viruses/nanoparticles has been carried out using FM[3,44,45]. However, with FM, information about the clustering states of intracellular particles could not be obtained. Plasmonic imaging are mainly used to investigate aggregation of nanoparticles in cells[37,46,47]. Correlative imaging using DFM/EM or FM/EM has been employed to investigate endocytosis of nanoparticles[19,48]. However, electron microscopy

can only provide information on internalization, aggregation and localization of nanoparticles in fixed cells. As best as we know, there has not been existing work reporting the movement of plasmonic nanoparticles in real time to study the effect of nanoparticle clustering on the intracellular transport. This work provides the first set of evidence supporting a direct correlation between clustering states of nanoparticles and their intracellular transport on the single-cell and single-particle level in real time, which provide important and valuable information on endocytosis of nanoparticles.

In conclusion, the present results demonstrate the use of dually emissive fPlas-gold nanoprobes for correlative imaging of their internalization and intracellular traffic in living cells. These fPlas-gold nanoparticles can be used as intrinsically emissive nanoprobes for long-term monitoring of the dynamics of endosome traffic in living cells without prior vesicle labelling. The cell entry of fPlas-gold was primarily in the monomeric form and occurred via multiple endocytic pathways. Real-time monitoring of fPlas-gold nanoparticle revealed arrays of motion types critically dependent on the clustering states of the nanoparticles during the intracellular transport. This study provides in-depth understanding of nanoparticle clustering and intracellular transport of clustered nanoparticles, which should open new opportunities for designing new drug delivery carriers or theranostic probes.

## Methods

**Synthesis and DNA modification of Au nanoparticles.** Au nanoparticles with diameter of 50 nm were synthesized by reducing $HAuCl_4$ with citrate in aqueous solution according to previous literature[49] and characterized using Ultraviolet–vis and TEM. dsDNA were modified on the surface of AuNPs by salt-ageing process (for detailed protocols see Supplementary Information).

**Cell culture.** HeLa cells were purchased from the Type Culture Collection of the Chinese Academy of Sciences (Shanghai, China). Cells were cultured in Minimum Essential Medium (MEM, GIBCO) with 10% fetal bovine serum (FBS, Invitrogen, Life Technologies) in a 5% $CO_2$ incubator at 37 °C. For imaging experiments, cells were seeded on 60 mm Petri dish (Corning) at a density of 300,000 and cultured overnight. Typically, HeLa cells were incubated with 0.1 nM fPlas-gold in MEM medium containing 10% FBS and washed with $1 \times$ PBS buffer (pH 7.4) before imaging. HeLa cell line used in this study is not on the ICLAC and NCBI biosample misidentified cell list. It was not authenticated and not tested for mycoplasma contamination.

**Confocal imaging.** Confocal images were recorded with a Leica TCS SP5 confocal microscope equipped with a live cell incubator and collected with a HC × PL APO $63 \times$, 1.4 NA oil-immersion objective. TIRF images were recorded with a Leica AM TIRF MC total internal reflection fluorescence microscope, and collected with a $100 \times$, 1.4 NA oil-immersion objective. fPlas-gold were excited with a 561 nm helium–neon laser, while the Chromeo 488 labelled tubulin (Goat anti-Mouse Chromeo 488 IgG (H&L), ab60313, Abcam), GFP fused tubulin (CellLight Tubulin-GFP, Life Technologies), LysoTracker Green dyes (LysoTracker Green DND-26, Life Technologies) and endosomes-GFP (CellLight Early Endosomes-GFP and Late Endosomes-GFP, BacMam 2.0, Life Technologies) were excited with a 488 nm Ar–Kr laser. Fluorescence images were analysed using Image J software. All the experiments were conducted at 37 °C unless otherwise mentioned.

**Correlation imaging.** Dark-field images and fluorescence images were recorded with an Olympus IX71 inverted microscope equipped with a 60/0.5–0.9 NA dry objective, a Olympus DP70 camera, a 100 W halogen lamp, a NKT supercontinuum laser and optical filters. Spectra were integrated for 10 s and collected with an Acton SP2300i spectrometer (Princeton Instruments). The optical paths of DFM and FM were independent from each other. DFM and FM images were recorded successively by altering operation mode, and analysed using MATLAB and Image J software, respectively. All the experiments were conducted at 37 °C unless otherwise mentioned.

**Quantification of intracellular fPlas-gold using ICP-AES.** HeLa cells were plated at a density of $3 \times 10^5$ cells per ml and cultured for 8 h. Then, the media were replaced with fresh media containing 0.1 nM fPlas-gold. After 0.5, 1, 2, 4 and 8 h incubation, the cells were washed with PBS three times, detached from the Petri

dish using trypsin − EDTA and collected. Cell pellets were digested with aqua regia ($HCl:HNO_3 = 3:1$) at room temperature overnight and the Au-197 content of resulting solution was measured with an Optima 8000 ICP-OES spectrometer (PerkinElmer). The average number of AuNPs internalized by one cell was determined according to the size of the Au nanosphere and the atomic weight of gold. Values represent mean ± s.e. from three independent experiments.

**Visualization of intracellular fPlas-gold using TEM.** Cells were fixed with 2.5% glutaraldehyde in $1 \times$ PBS buffer (pH 7.4) and stained with 1% $OsO_4$ at 4 °C. After gradual dehydration with ethanol and acetone, cell pellets were embedded in Epon 812 resins (Electron Microscopy Science) and sliced to pieces with a thickness of 70 nm then stained with uranyl acetate. Images of cell slices were taken with a FEI Tecnei G2-205 Twin TEM using a beam voltage of 80 kV.

**Data availability.** Data supporting the findings of this study are available within the article (and its Supplementary Information files) and from the corresponding authors upon reasonable request.

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

## Acknowledgements

This work was supported by the National Natural Science Foundation of China (21390414, 21227804, 21329501, 31470970, 21505148), the Ministry of Science and Technology of China (2013CB933802, 2013CB932803, 2016YFA0201200), the Chinese Academy of Sciences (QYZDJ-SSW-SLH031), the Nanotechnology Program of Shanghai Science and Technology Committee (13NM1402300), and the Natural Science Foundation of Shanghai (15ZR1448400, 15ZR1448700). This project was supported in part by the National Institute on Aging of National Institutes of Health (Grant AG028709).

## Author contributions

C.F. and R.L. conceived the study. M.L., Q.L., L.L. and C.F. designed experiments. M.L., Q.L. and L.L. performed experiments. J.L. (Jiang Li) assisted confocal imaging. K.W. and J.L.(Jiajun Li) assisted DFM imaging. M.L. (Min Lv), N.C. and H.S. assisted cellular experiments. J. S., L.W. J.L. (Joon Lee) and R.L. assisted TEM imaging. M.L., Q.L., L.L., J.S., R.L. and C.F. analysed data and wrote the paper.

## Additional information

**Competing interests:** The authors declare no competing financial interests.

