## [Peer Review File · Nature Communications]

Reviewer #1 (Remarks to the Author):

This manuscript describes a novel application of dye-labeled DNA-conjugated AuNPs in visualizing intracellular transport within living cells. The most important result presented involved the correlative imaging of the internalization and intracellular trafficking of the nanoconjugates using both fluorescence microscopy (FM, for the DNA) and dark-field microscopy (DFM, for the gold nanoparticles). It is a fine contribution that could be published in Nature Communications.

Other points:

1. Choice of dye and presence of DNA:

a. The authors claim that the DNA-AuNPs remain intact; however, there is low colocalization of the dye and the AuNPs primarily due to the photobleaching or the low quantum yield of Cy3. The authors should use other dyes with a similar range, such as TAMRA or rhodamine, that have better quantum yields and that are less sensitive to photobleaching.

b. The Cy3 dye is present on the DNA and not on the nanoparticle. It has been shown that DNA-functionalized AuNPs are susceptible to degradation in cells (J. Am. Chem. Soc., 2014, 136 (21), pp 7726-7733). Hence, the presence of the dye in the endosome cannot be directly correlated to the presence of the whole conjugate, especially during endosomal fusion to form late endosomes.

c. The authors should report the Mander's colocalization coefficients between the DNA and the endosomal stains.

d. The FM data needs to be more thoroughly analyzed and better corroborated with the DFM imaging data for this to be considered a correlative technique.

Reviewer #2 (Remarks to the Author):

In the current work, the authors investigate internalization and intracellular traffic of a DNA-mediated dually emissive gold probe in living cells. The use of both plasmonic and fluorescent properties of the particles for monitoring intra-cellular processes is certainly a powerful method, which opens new advances in bio-imaging. However, one of the key prerequisites of correct interpretation of the results is monodispersity of the probe. However, supplementary figure 1(b) clearly shows that the gold particles differ in size. Moreover, even by inspecting the provided low resolution image one can see that size of some of the particles differ by a factor of 2. This raises the question on interpretation of the spectra shown in supplementary figure 3 (b) and (c). The clear deviation of the measured data from the theoretical model is most likely caused by the broad size distribution of the particles. This makes impossible to make any reliable conclusions from the data. How do the authors distinguish clustering of the particles from the variation of their size?

Additional minor points:

1. In my view, the manuscript contains excess of acronyms, which are so not commonly used in the fluorescence microscopy community. For instance, SP, LC, MSD, SC, ... This makes the manuscript harder to read. The authors could especially avoid overload of acronyms in the figure captions.

2. Figure 1(d): why do the authors use a 3d representation of the plot? The plot can be shown exactly in the same way as the one in figure (e), which would clearly improve its readability.

After taking into account the above points, the results may certainly be of interest for the researchers working in the related fields. However, I believe that even in a revised form, the manuscript is more appropriate for specialized fluorescence microscopy and bio-imaging journals

rather than Nature Communications.

Reviewer #3 (Remarks to the Author):

The manuscript by Lal & Fan et al. applied gold nanoparticles, which have the distinct ability to change spectra under different clustering status, to investigate nanoparticle clustering in the endocytosis process of living cells; also, by combining dark field microscopy, fluorescent imaging and electron microscopy, this manuscript carried out an in-depth study of the endocytosis-based transport of DNA-decorated gold nanoparticles. This Reviewer believes that this work is both novel and important. The novelty and importance of this work mainly come from: 1) although the phenomenon of spectrum change of gold nanoparticles upon aggregation is well-known and it has been used to study nanoparticle clustering before, its application for studying nanoparticle clustering in living cells is new, and this effort is of significant value considering the fact that nanoparticle clustering in living cells is highly important for nanobiotechnology and nanomedicine and yet is lack of careful studies. 2) Some of the findings are of great interests: after gold nanoparticles are internalized into cells, inside early endosomes are mainly single particles, and these particles are transported along microtubules to the perinuclear region mainly in single particle status; nanoparticle clustering occurs mainly in later stages. The experiments are well-designed and the presentation is clear. Thus, this Reviewer recommends Acceptance of this manuscript provided that the following specific points are properly addressed:

- 1) In Figure 1, description of (f) appears to be missing in the figure legend.
- 2) In Figure 2, "microtubulin" should be "microtubule".
- 3) In the method to study colocalization, how are tMr values determined?
- 4) The proposed explanation on why the transport speed for gold nanoparticles as the cargo compared with that for a "normal" cargo was that the density of gold is higher than that of a "normal" cargo. This explanation is not convincing. In the cargo transport in living cells, gravity is not a main resistance to overcome. Rather, the main resistance comes from drag force (hydrodynamic friction due to viscosity), which is mainly related to the size and shape of the cargo, and is independent of the density. Take the 2-dimensional movement of cargo imaged and tracked in this manuscript as an example, the direction of gravity is in the direction perpendicular to the direction of cargo movement, and thus shouldn't cause significant resistance.
- 5) In Line 141, "nuclear region" should be "perinuclear region", as the particles didn't really enter the cell nucleus.
- 6) In Line 25, "know" should be "known".
- 7) In Line 257, "vascular" should be "vesicle".

Reviewer #4 (Remarks to the Author):

The paper describes an application of fluorescently labeled dsDNA-coated gold nanoparticles in studies of nanoparticle cell entry pathways and intracellular trafficking. The most interesting results are associated with observations of different motion patterns in cells using dark-field microscopy; this approach can potentially allow a quantitative statistical analysis of interactions between different endosomal vesicles. However, the authors have stopped short of carrying out such quantitative studies limiting their experiments to a set of observations/examples. Therefore, the paper is an interesting extension/continuation of work that has been previously done by other groups (representative studies are referenced by the authors). However, it does not have sufficient novelty to warrant publication in Nature Communications and it would be better suited for a specialty journal in the field of nanotechnology or imaging. Furthermore, there are multiple technical concerns that need to be addressed which are summarized below.

Major Concerns deal with the validation of nanoparticles association with various endosomal compartments in cells:

1) All "spots" in dark-field microscopy (DFM) are categorized as single particles (SP), small clusters (SC) and large clusters (LC) solely based on their apparent color - green, yellow and red, respectively. How were these colors identified? In other words, how it was determined that a color is green, yellow or red. This is an important definition that has to be done in a quantitative way because many conclusions in the paper are based on it.

2) The authors have not fully explored the double contrast capability of their nanoparticles - fluorescent and DMF imaging have been carried out separately. The problem is that the authors have never established that their SP, SC and LC particles are actually associated with early endosomes, late endosomes and lysosomes, respectively, as it is claimed in data analysis/results. This correlation was assumed based on fluorescent co-localization measurements that have been carried out separately from DMF. Furthermore, there is a problem in interpretation of the co-localization studies in Figure 2. It appears from the images in Fig. 2c that nanoparticles (red fluorescence) are not co-localized with various endosomal compartments (green fluorescence) as it is claimed because there is no evident yellow color in the overlay images that would be an indication of a co-localization.

3) Fig. 3: similarly to the concern above, it appears that the correlation of stage I - low motility particles, stage 2 - high motility particles, and stage 3 - low motility particles with those bound to the cellular membrane, in early endosomes, and in lysosomes, respectively, is also assumed because there is no co-labeling of the endosomal compartments in these experiments.

Minor comments:

1) Suppl Fig 1: an increase in the absorbance of particle conjugates above 600nm can be indicative of some aggregation. TEM data cannot be used to evaluate particle aggregation state due to well-known sample preparation artifacts. The particles should be characterized by dynamic light scattering (DLS). In addition, Z-potential should be also determined because particle charge is critically important in interactions with cells.

2) The statement (lines 126-127) - "Typically, we could easily distinguish SP (n=1; green), small clusters (SC; n=2-5; yellow) and large clusters (LC; n>5; bright yellow) under DFM" - is not generally true because the color is also a function of changes in microenvironment surrounding gold nanoparticles that also needs to be taken into account.

3) tMr data analysis should be described.

4) In Fig. 2d, it appears that only a fraction of nanoparticles is located in endosomes at any point of time. Where are the rest of the nanoparticles?

5) Fig. 5: is there statistically significant difference between data for SP, SC and LC?

Responses to Reviewer #1:

This manuscript describes a novel application of dye-labeled DNA-conjugated AuNPs in visualizing intracellular transport within living cells. The most important result presented involved the correlative imaging of the internalization and intracellular trafficking of the nanoconjugates using both fluorescence microscopy (FM, for the DNA) and dark-field microscopy (DFM, for the gold nanoparticles). It is a fine contribution that could be published in Nature Communications. Other points:

Concern 1:

Choice of dye and presence of DNA: a. The authors claim that the DNA-AuNPs remain intact; however, there is low colocalization of the dye and the AuNPs primarily due to the photobleaching or the low quantum yield of Cy3. The authors should use other dyes with a similar range, such as TAMRA or rhodamine, that have better quantum yields and that are less sensitive to photobleaching.

Response:

We highly appreciate the comments and suggestions from the Reviewer 1. We have performed additional experiments to improve our manuscript as this reviewer suggested.

Regarding the choice of Cy3, we did try several types of fluorophores. We found that Cy3 works fine in most situations. However, when we used TAMRA for our experiments, we obtained poorer image quality. This might be caused by the higher hydrophobic properties of TAMRA, leading to fPlas-gold aggregation. Here we include these data for reviewer 1, as shown below. The left figure is a representative DFM image of HeLa cells incubated with TAMRA-dsDNA-AuNPs for 4 h, in which much more of the intracellular AuNPs existed as large clusters compared to Cy3-dsDNA-AuNPs (right).

Concern 2:

The Cy3 dye is present on the DNA and not on the nanoparticle. It has been shown that DNA-functionalized AuNPs are susceptible to degradation in cells (J. Am. Chem. Soc., 2014, 136 (21), pp 7726-7733). Hence, the presence of the dye in the endosome cannot be directly correlated to the presence of the whole conjugate, especially during endosomal fusion to form late endosomes.

Response:

We thank the Reviewer 1 for this constructive suggestion. The experimental system in our manuscript is different from the paper mentioned by the reviewer 1 (J. Am. Chem. Soc., 2014, 136 (21), pp 7726-7733). In that work, the authors investigated the fluorescence of Cy5-ssDNA-functionalized quantum dots (QDs of ~9 nm) in C166 cells, rather than ssDNA-AuNPs.

In our study, we observed Cy3 fluorescence and DFM signal of 50 nm AuNPs in HeLa cells within 24 h and did not find obvious signal separation. We noticed some dots present in DFM were missing in FM, which might arise from the photobleaching of Cy3, insufficient sensitivity of FM and/or disassembly of f-Plas gold. Importantly, we did not observe Cy3 fluorescence unattached to AuNPs, indicating that the disassembled Cy3-DNA cannot be visualized in FM and thus would not influence our data analysis. These data have been included in our manuscript (P5 line 26-29, Figure S6).

Supplementary Figure 6. Time-dependent cellular uptake of fPlas-gold. Representative examples of DFM images of HeLa cells incubated with 0.1 nM fPlas-gold for different time. Both DFM images and FM images were taken by using the correlative microscope.

Concern 3:

The authors should report the Mander's colocalization coefficients between the DNA and the endosomal stains.

Response:

We thank the Reviewer 1 for this suggestion. The tMr values that we obtained using

imageJ software are exactly the thresholded Mander's coefficients. We have made this point clearer in our manuscript with the following line added on page 7 and cited the literature (reference 42, Manders, E.M.M., Verbeek, F.J., Aten, J.A. Measurement of co-localization of objects in dualcolor confocal images. Journal of Microscopy 169: 375-382 (1993)):

“Time-lapse studies revealed the degree of co-localization of fPlas-gold with these cellular compartments, which were quantified by the thresholded Mander's colocalization coefficients (tMr)⁴² representing the percentage of the Cy3 fluorescence overlaid with the GFP.”

Concern 4:

The FM data needs to be more thoroughly analyzed and better corroborated with the DFM imaging data for this to be considered a correlative technique.

Response:

We highly appreciate this suggestion of the Reviewer 1 and have performed additional experiments to further prove the correlation of FM data and DFM data.

i) We detected Cy3 fluorescence and DFM signals of AuNPs of the same field of view using the combined microscope and did not find signal separation within 24 h, which strongly suggested the correlation of fluorescence signal and DFM signal. We have added the following line on page 5 in our manuscript and these data have been included as Supplementary Figure 6.

“Importantly, we did not observe Cy3 fluorescence unattached to AuNPs or obvious separation of Cy3 fluorescence and AuNPs signal, indicating the disassembled Cy3-DNA could not be visualized in FM and thus would not influence our data analysis (Supplementary Figure 6).”

Supplementary Figure 6. Time-dependent cellular uptake of fPlas-gold. Representative examples of DFM images of HeLa cells incubated with 0.1 nM fPlas-gold for different time. Both DFM images and FM images were taken by using the correlative microscope.

ii) We overlaid DFM and FM signals of f-Plas gold with FM signals of endosomes in fixed HeLa cells and found nice colocalization of them, as shown in the figures below. These data have been included in our manuscript in Supplementary Figure 10 and Supplementary Figure 11.

Supplementary Figure 10 Colocalization of fPlas-gold with GFP-fused early

endosomes as revealed with DFM and FM imaging. (a) DFM image of fPlas-gold; FM image of (b) fPlas- gold and (c) early endosomes; (d) overlay of a and b; (e) overlay of a and c; (e) overlay of b and c. To avoid influence of green GFP fluorescence on DFM signal of green spots, the green fluorescence was converted to blue color in e. Both DFM images and FM images of fPlas-gold were taken on the correlative microscope. FM images of GFP-fused early endosomes were taken on confocal microscope layer by layer and then reconstructed.

Supplementary Figure 11 Colocalization of fPlas-gold with GFP-fused late endosomes as revealed with DFM and FM imaging. (a) DFM image of fPlas-gold; FM image of (b) fPlas-gold and (c) late endosomes; (d) overlay of a and b; (e) overlay of a and c; (e) overlay of b and c. To avoid influence of green GFP fluorescence on DFM signal of green spots, the green fluorescence was converted to blue color in e. Both DFM images and FM images of fPlas-gold were taken on the correlative microscope. FM images of GFP-fused late endosomes were taken on confocal microscope layer by layer and then reconstructed.

Responses to Reviewer #2:

Concern 1:

In the current work, the authors investigate internalization and intracellular traffic of a DNA-mediated dually emissive gold probe in living cells. The use of both plasmonic and fluorescent properties of the particles for monitoring intra-cellular processes is certainly a powerful method, which opens new advances in bio-imaging. However, one of the key prerequisites of correct interpretation of the results is monodispersity of the probe. However, supplementary figure 1(b) clearly shows that the gold particles differ in size. Moreover, even by inspecting the provided low resolution image one can see that size of some of the particles differ by a factor of 2. This raises the question on interpretation of the spectra shown in supplementary figure 3 (b) and (c). The clear deviation of the measured data from the theoretical model is most likely caused by the broad size distribution of the particles. This makes impossible to make any reliable conclusions from the data. How do the authors distinguish clustering of the particles from the variation of their size?

Response:

We highly appreciate the comments and suggestions from the Reviewer 2. We have performed additional experiments and improved our data.

The size deviation of AuNPs would not cause change of optical properties under DFM imaging. Additional data has been included in Supplementary Figure 1c and 1d, revealing that over 95% of AuNPs represents as green spots, based on the analysis of over 1,000 randomly selected particles.

Supplementary Figure 1. Characterization of f-Plas gold. (a) Nanoparticle absorbance spectra before (black) and after (red) DNA modification. (b) A representative TEM image and (c) three DFM image of fPlas-gold. (d) Over 1000 dots in DFM images were analyzed, indicating more than 95% of fPlas-gold existed as green dots in DFM images.

Concern 2:

In my view, the manuscript contains excess of acronyms, which are so not commonly used in the fluorescence microscopy community. For instance, SP, LC, MSD, SC, ... This makes the manuscript harder to read. The authors could especially avoid overload of acronyms in the figure captions.

Response:

We apologize for using excess of acronyms and making the manuscript hard to read. We have reduced acronyms in figure captions.

Concern 3.

Figure 1(d): why do the authors use a 3d representation of the plot? The plot can be shown exactly in the same way as the one in figure (e), which would clearly improve its readability.

Response:

We have replaced Figure 1d as the reviewer suggested.

Responses to Reviewer #3:

The manuscript by Lal & Fan et al. applied gold nanoparticles, which have the distinct ability to change spectra under different clustering status, to investigate nanoparticle clustering in the endocytosis process of living cells; also, by combining dark field microscopy, fluorescent imaging and electron microscopy, this manuscript carried out an in-depth study of the endocytosis-based transport of DNA-decorated gold nanoparticles. This Reviewer believes that this work is both novel and important. The novelty and importance of this work mainly come from: 1) although the phenomenon of spectrum change of gold nanoparticles upon aggregation is well-known and it has been used to study nanoparticle clustering before, its application for studying nanoparticle clustering in living cells is new, and this effort is of significant value considering the fact that nanoparticle clustering in living cells is highly important for nanobiotechnology and nanomedicine and yet is lack of careful studies. 2) Some of the findings are of great interests: after gold nanoparticles are internalized into cells, inside early endosomes are mainly single particles, and these particles are transported along microtubules to the perinuclear region mainly in single particle status; nanoparticle clustering occurs mainly in later stages. The experiments are well-designed and the presentation is clear. Thus, this Reviewer recommends Acceptance of this manuscript provided that the following specific points are properly addressed:

Concern 1:

In Figure 1, description of (f) appears to be missing in the figure legend.

Response:

Thank you for this constructive suggestion. Actually, the description of (f) follows the one of (b).

(b) DFM and (f) TEM images for time-evolution of fPlas-gold incubated with Hela cells.

Concern2:

In Figure 2, "microtubulin" should be "microtubule".

Response:

Thanks for this suggestion. We have modified it in the manuscript as suggested.

Concern3:

In the method to study colocalization, how are tMr values determined?

Response:

The tMr values were obtained using imageJ software. The values were exactly the thresholded Mander's coefficients. We have made this point clearer in our manuscript with the following line added on page 7 and cited the literature (reference 42, Manders, E.M.M., Verbeek, F.J., Aten, J.A. Measurement of co-localization of objects in dualcolor confocal images. Journal of Microscopy 169: 375-382 (1993)):

“Time-lapse studies revealed the degree of co-localization of fPlas-gold with these cellular compartments, which were quantified by the thresholded Mander's colocalization coefficients (tMr)⁴² representing the percentage of the Cy3 fluorescence overlaid with the GFP.”

Concern 4:

The proposed explanation on why the transport speed for gold nanoparticles as the cargo compared with that for a "normal" cargo was that the density of gold is higher than that of a "normal" cargo. This explanation is not convincing. In the cargo transport in living cells, gravity is not a main resistance to overcome. Rather, the main resistance comes from drag force (hydrodynamic friction due to viscosity), which is mainly related to the size and shape of the cargo, and is independent of the density. Take the 2-dimensional movement of cargo imaged and tracked in this manuscript as an example, the direction of gravity is in the direction perpendicular to the direction of cargo movement, and thus shouldn't cause significant resistance.

Response:

We appreciate this suggestion very much. We further analyzed our TEM data and found out the vesicle size increased along with the numbers of encapsulated AuNPs (Figure S20). The numbers of encapsulated AuNPs by early endosomes were the smallest and the one by lysosomes were the biggest; while the ones captured by late endosomes are in between, which is in agreement of the size of these three organelles (early endosomes < late endosomes < lysosomes).

Supplementary Figure 20. Dependence of vesicle sizes and the number of encapsulated AuNPs. TEM images of randomly selected (a) ten vesicles containing

single particles and (b) ten vesicles containing small clusters. (c) The measured vesical diameters of single particles and small clusters. Data were presented as the mean \pm SEM. **P<0.01, according to student-t test.

Furthermore, we examined the dynamics of fPlas-gold transport and movement of early endosomes, late endosomes and lysosomes (Supplementary Figure 13 and Supplementary Video 6a-c) and found out that most of the fPlas-gold captured in early endosomes showed high mobility and most of the ones in late endosomes and lysosomes showed low mobility.

Supplementary Figure 13. Representative trajectories of intracellular fPlas-gold of high mobility (magenta) and low mobility (blue) in FM images.

Taken together, we conclude that the moving speed of vesicles is highly dependent on the vesicle size, as the reviewer suggested. We have included these data and provided short discussion in our manuscript as below. **We also deleted the discussion on the effect of overloaded vesicles on their speed, as suggested by this reviewer.**

On Page 8, “**We then examined the dynamics of fPlas-gold transport and movement of these three organelles (Figure S13 and Supplementary Video 2: $\Delta t = 3$ s, total time = 90 s for a and b, 450s for c) and the observed results were in good consistence with our proposal, that most of the fPlas-gold captured in early endosomes showed high mobility and most of the ones in late endosomes and lysosomes showed low mobility.**”

On page 11, “**Notably, TEM results demonstrated that the vesicle size increased with numbers of single AuNPs captured in it (Figure S20). The numbers of AuNPs in these three organelles were in consistence with their sizes.**”

Concern 5:

In Line 141, "nuclear region" should be "perinuclear region", as the particles didn't really enter the cell nucleus.

Response:

Thanks for this suggestion. We have modified it as suggested.

Concern 6:

In Line 25, "know" should be "known".

Response:

Thanks for this suggestion. We have modified it as suggested.

Concern 7:

In Line 257, "vascular" should be "vesicle".

Response:

Thanks for this suggestion. We have modified it as suggested.

Responses to Reviewer #4:

The paper describes an application of fluorescently labeled dsDNA-coated gold nanoparticles in studies of nanoparticle cell entry pathways and intracellular trafficking. The most interesting results are associated with observations of different motion patterns in cells using dark-field microscopy; this approach can potentially allow a quantitative statistical analysis of interactions between different endosomal vesicles. However, the authors have stopped short of carrying out such quantitative studies limiting their experiments to a set of observations/examples. Therefore, the paper is an interesting extension/continuation of work that has been previously done by other groups (representative studies are referenced by the authors). However, it does not have sufficient novelty to warrant publication in Nature Communications and it would be better suited for a specialty journal in the field of nanotechnology or imaging. Furthermore, there are multiple technical concerns that need to be addressed which are summarized below.

Concern 1:

Major Concerns deal with the validation of nanoparticles association with various endosomal compartments in cells:

All "spots" in dark-field microscopy (DFM) are categorized as single particles (SP), small clusters (SC) and large clusters (LC) solely based on their apparent color - green, yellow and red, respectively. How were these colors identified? In other words, how it was determined that a color is green, yellow or red. This is an important definition that has to be done in a quantitative way because many conclusions in the paper are based on it.

Response:

We first identified closely packed fPlas-gold clusters with SEM, and then performed ex-situ DFM to image individual clusters in the same view. After analyzing DFM data of these clusters, we found that a single particle (SP) of fPlas-gold exhibited green color, whereas the color of clustered fPlas-gold gradually turned to yellow along with the increased number (n) of particles (from n=1 to n=10) (Supplementary Figure 3a). We then performed finite-difference time-domain (FDTD) simulation and found out the simulated scattering spectra generally consistent with the measured ones. Taking all of these results together, we concluded the characteristic colors (green, yellow and bright yellow in DFM images) and scattering wavelengths of differentially sized clusters were resulted from the plasmonic coupling in fPlas-gold clusters.

We also compared our naked-eye classification of randomly-selected 20 fPlas-gold clusters with the results based on scattering spectra, and found a good agreement. Data is shown below.

Concern 2:

The authors have not fully explored the double contrast capability of their nanoparticles - fluorescent and DMF imaging have been carried out separately. The problem is that the authors have never established that their SP, SC and LC particles are actually associated with early endosomes, late endosomes and lysosomes, respectively, as it is claimed in data analysis/results. This correlation was assumed based on fluorescent co-localization measurements that have been carried out separately from DMF. Furthermore, there is a problem in interpretation of the co-localization studies in Figure 2. It appears from the images in Fig. 2c that nanoparticles (red fluorescence) are not co-localized with various endosomal compartments (green fluorescence) as it is claimed because there is no evident yellow color in the overlay images that would be an indication of a co-localization.

Response:

We thank this reviewer for these constructive comments. We have improved the figure quality of our Figure 2c. We also performed additional experiments to further confirm our conclusions. Importantly, we have demonstrated the correlation of fluorescence and DFM signals in this revision as shown below.

First, we observed Cy3 fluorescence and DFM signal of 50 nm AuNPs in HeLa cells within 24 h and did not find obvious signal separation, which strongly suggested the correlation of fluorescence signal and DFM signal. These data have been included in our manuscript (P5 line 26-29, Figure S6).

Supplementary Figure 6. Time dependent cellular uptake of fPlas-gold. Representative examples of DFM images of HeLa cells incubated with 0.1 nM fPlas-gold for different time. Both DFM images and FM images of fPlas-gold were taken on the correlative microscope.

Second, we overlaid DFM and FM signals of fPlas-gold with FM signals of endosomes in fixed HeLa cells and found nice colocalization of them, as shown in the figures below. These data have been included in our manuscript in Figure S10 and 11.

Supplementary Figure 10. Colocalization of fPlas-gold with GFP-fused early endosomes revealed by DFM and FM: (a) DFM image of fPlas-gold; FM image of (b) fPlas-gold and (c) early endosomes; (d) overlay of a and b; (e) overlay of a and c; (e) overlay of b and c. To avoid influence of green GFP fluorescence on DFM signal of green spots, the green fluorescence was converted to blue color in e. Both DFM images and FM images of fPlas-gold were taken on the correlative microscope. FM

images of GFP-fused early endosomes were taken on confocal microscope layer by layer and then reconstructed.

Supplementary Figure 11. Colocalization of fPlas-gold with GFP-fused late endosomes revealed by DFM and FM: (a) DFM image of fPlas-gold; FM image of (b) fPlas-gold and (c) late endosomes; (d) overlay of a and b; (e) overlay of a and c; (e) overlay of b and c. To avoid influence of green GFP fluorescence on DFM signal of green spots, the green fluorescence was converted to blue color in e. Both DFM images and FM images of fPlas-gold were taken on the correlative microscope. FM images of GFP-fused late endosomes were taken on confocal microscope layer by layer and then reconstructed.

Third, the movement of fPlas-gold and early endosomes, late endosomes and lysosomes in living cells were investigated, respectively, by detecting Cy3 fluorescence of f-Plas gold and fluorescence of these stained intracellular organelles. The observed results demonstrated that f-Plas gold and organelles move together, which strongly suggested capture of f-Plas gold by endosomes and lysosomes. These results were included in Supplementary Videos 6a-c and Supplementary Figures 13.

Supplementary Figure 13. Representative trajectories of intracellular f-Plas gold of high mobility (magenta) and low mobility (blue) in confocal FM images.

In short summary, we observed single particles, small clusters and large clusters

captured in early endosomes, late endosomes and lysosomes, respectively, using TEM. We also observed colocalization of Cy3 fluorescence with GFP fluorescence of endosomes and lysotracker signal of lysosomes in fixed cells, and movement of Cy3 fluorescence with fluorescence of endosomes and lysosomes in living cells. Using DFM, we observed movement of single fPlas-gold and found out the movement of single particles, small clusters and large clusters were different. Taken all these data together, we believe we obtain new knowledge of nanoparticle entry, clustering and transport by using this new method.

Concern 3:

Fig. 3: similarly to the concern above, it appears that the correlation of stage I - low motility particles, stage 2 - high motility particles, and stage 3 - low motility particles with those bound to the cellular membrane, in early endosomes, and in lysosomes, respectively, is also assumed because there is no co-labeling of the endosomal compartments in these experiments.

Response:

We thank the reviewer for this suggestion. We performed additional experiments to investigate the movement of fPlas-gold and early endosomes, late endosomes and lysosomes in living cells, respectively, by detecting Cy3 fluorescence of fPlas-gold and fluorescence of these stained intracellular organelles. As shown in Supplementary Videos 6 a-c and Supplementary Figures 13, most of the fPlas-gold captured in early endosomes showed high mobility, and most of the ones in late endosomes and lysosomes showed low mobility, which is in nice agreement with the data shown in Figure 3. We have included this in our manuscript on Page 8 as below.

“We then examined the dynamics of fPlas-gold transport and movement of these three organelles (Figure S13 and Supplementary Video 2: $\Delta t = 3$ s, total time = 90 s for a and b, 450s for c) and the observed results were in good consistence with our proposal, that most of the fPlas-gold captured in early endosomes showed high mobility and most of the ones in late endosomes and lysosomes showed low mobility.”

Supplementary Figure 13. Representative trajectories of intracellular f-Plas gold of high mobility (magenta) and low mobility (blue) in confocal FM images.

Minor comments:

Concern 4:

Suppl Fig 1: an increase in the absorbance of particle conjugates above 600nm can be indicative of some aggregation. TEM data cannot be used to evaluate particle aggregation state due to well-known sample preparation artifacts. The particles should be characterized by dynamic light scattering (DLS). In addition, Z-potential should be also determined because particle charge is critically important in interactions with cells.

Response:

We have included data of DLS and Z-potential in our manuscript on page 4 as follows.

“Compared to unmodified AuNPs, Z-potential of assembled fPlas-gold decreased from -44.0 ± 0.4 mV to -34.4 ± 0.4 mV, and the value of dynamic light scattering (DLS) increased from 46.3 ± 0.5 nm to 71.6 ± 1.8 nm.”

Regarding the aggregation of AuNPs, we first picked up different AuNPs clusters using SEM and then observed them using DFM. After analyzing DFM data of these known aggregated clusters of different aggregation states, we defined AuNPs cluster into three main classes based on the observed light scattering spectra.

Concern 5:

The statement (lines 126-127) - "Typically, we could easily distinguish SP (n=1; green), small clusters (SC; n=2-5; yellow) and large clusters (LC; n>5; bright yellow) under DFM" - is not generally true because the color is also a function of changes in microenvironment surrounding gold nanoparticles that also needs to be taken into account.

Response:

Based on our theoretical calculation, the cell microenvironment should not influence the optical properties of fPlas-gold. This is further confirmed by our experimental observation that extra- and intra-cellular fPlas-gold exhibit same optical scattering properties. We picked scattering spectrum of the extracellular green dot in Figure S3a and the intracellular green dot in Figure S4a, and overlaid them together. As shown in those two figures below, the scattering spectra of the extracellular SP (black) overlaid with the one of intracellular SP (red) perfectly. Therefore, the cellular microenvironment did not affect the optical scattering properties (i.e. color) of fPlas-gold. These data have been included in manuscript (P5 line 10-13 and Figure S5).

Supplementary Figure 5. Comparison of light scattering spectra of extracellular and intracellular fPlas-gold. The scattering spectra of the green dot in Figure S3a and S4a overlaid perfectly with each other.

Concern 6:

tMr data analysis should be described.

Response:

We thank reviewer 4 for this suggestion. The tMr values we obtained using imageJ software are exactly the thresholded Mander's coefficients. We have made this point clearer in our manuscript with the following line added on page 7 and cited the literature (reference 42, Manders, E.M.M., Verbeek, F.J., Aten, J.A. Measurement of co-localization of objects in dual color confocal images. Journal of Microscopy 169: 375-382 (1993)):

“Time-lapse studies revealed the degree of co-localization of fPlas-gold with these cellular compartments, which were quantified by the thresholded Mander's colocalization coefficients (tMr)⁴² representing the percentage of the Cy3 fluorescence overlaid with the GFP.”

Concern 7:

In Fig. 2d, it appears that only a fraction of nanoparticles is located in endosomes at any point of time. Where are the rest of the nanoparticles?

Response:

We have improved the quality of Figure 2d, which should give a better demonstration of colocalization of fPlas-gold with endosomes and lysosomes. Moreover, multi-pathways were involved in the cellular uptake process of fPlas-gold and thus some of the NPs would be located in other organelles which were not stained.

Concern 8:

Fig. 5: is there statistically significant difference between data for SP, SC and LC?

Response:

Thanks for this suggestion. We have added P value calculated by student-t test in Figure 5.

Responses to Reviewer #1:

This manuscript describes a novel application of dye-labeled DNA-conjugated AuNPs in visualizing intracellular transport within living cells. The most important result presented involved the correlative imaging of the internalization and intracellular trafficking of the nanoconjugates using both fluorescence microscopy (FM, for the DNA) and dark-field microscopy (DFM, for the gold nanoparticles). It is a fine contribution that could be published in Nature Communications. Other points:

Concern 1:

Choice of dye and presence of DNA: a. The authors claim that the DNA-AuNPs remain intact; however, there is low colocalization of the dye and the AuNPs primarily due to the photobleaching or the low quantum yield of Cy3. The authors should use other dyes with a similar range, such as TAMRA or rhodamine, that have better quantum yields and that are less sensitive to photobleaching.

Response:

We highly appreciate the comments and suggestions from the Reviewer 1. We have performed additional experiments to improve our manuscript as this reviewer suggested.

Regarding the choice of Cy3, we did try several types of fluorophores. We found that Cy3 works fine in most situations. However, when we used TAMRA for our experiments, we obtained poorer image quality. This might be caused by the higher hydrophobic properties of TAMRA, leading to fPlas-gold aggregation. Here we include these data for reviewer 1, as shown below. The left figure is a representative DFM image of HeLa cells incubated with TAMRA-dsDNA-AuNPs for 4 h, in which much more of the intracellular AuNPs existed as large clusters compared to Cy3-dsDNA-AuNPs (right). Since this is a negative control, we did not include it in the manuscript.

TAMRA

Cy3

Concern 2:

The Cy3 dye is present on the DNA and not on the nanoparticle. It has been shown that DNA-functionalized AuNPs are susceptible to degradation in cells (J. Am. Chem.

Soc., 2014, 136 (21), pp 7726-7733). Hence, the presence of the dye in the endosome cannot be directly correlated to the presence of the whole conjugate, especially during endosomal fusion to form late endosomes.

Response: We thank the Reviewer 1 for this constructive suggestion. The experimental system in our manuscript is different from the paper mentioned by the reviewer 1 (J. Am. Chem. Soc., 2014, 136 (21), pp 7726-7733). In that work, the authors investigated the fluorescence of Cy5-ssDNA-functionalized quantum dots (QDs of ~9 nm) in C166 cells, rather than ssDNA-AuNPs.

In our study, we observed Cy3 fluorescence and DFM signal of 50-nm AuNPs in HeLa cells within 24 h and did not find obvious signal separation. We noticed some dots present in DFM were missing in FM, which might arise from the photobleaching of Cy3, insufficient sensitivity of FM and/or disassembly of fPlas-gold. Importantly, we did not observe Cy3 fluorescence unattached to AuNPs, indicating that the disassembled Cy3-DNA cannot be visualized in FM and thus would not influence our data analysis. These data have been included in our manuscript (**P6, Figure S7**).

Supplementary Figure 7. Time-dependent cellular uptake of fPlas-gold. Representative examples

of DFM images of HeLa cells incubated with 0.1 nM fPlas- gold for different time. Both DFM images and FM images were taken by using the correlative microscope.

Concern 3:

The authors should report the Mander's colocalization coefficients between the DNA and the endosomal stains.

Response:

We thank the Reviewer 1 for this suggestion. The tMr values that we obtained using imageJ software are exactly the thresholded Mander's coefficients. We have made this point clearer in the revised manuscript (**P7**) and cited a new reference (**Ref 42**).

P7: “Time-lapse studies revealed the degree of co-localization of fPlas-gold with these cellular compartments, which were quantified by the thresholded Mander's colocalization coefficients (tMr)⁴² representing the percentage of the Cy3 fluorescence overlaid with the GFP.”

Concern 4:

The FM data needs to be more thoroughly analyzed and better corroborated with the DFM imaging data for this to be considered a correlative technique.

Response:

We highly appreciate this suggestion of the Reviewer 1 and have performed additional experiments to better establish the correlation of FM data and DFM data.

1) We detected Cy3 fluorescence and DFM signals of AuNPs of the same field of view using the combined microscope and did not find signal separation within 24 h, which strongly suggested the correlation of fluorescence signal and DFM signal. We have added the following discussion in the revised manuscript (**P5, and Supplementary Figure 7**).

“Importantly, we did not observe Cy3 fluorescence detached from AuNPs or apparent separation of Cy3 fluorescence and AuNPs signal, implying that the observed fluorescence signal in FM results from the assembled fPlas-gold (Supplementary Figure 7).”

Supplementary Figure 7. Time-dependent cellular uptake of fPlas-gold. Representative examples of DFM images of HeLa cells incubated with 0.1 nM fPlas-gold for different time. Both DFM images and FM images were taken by using the correlative microscope.

2) We overlaid DFM and FM signals of f-Plas gold with FM signals of endosomes in fixed HeLa cells and found nice colocalization of them, as shown in the figures below. These data have been included in the revised manuscript (**Supplementary Figure 11 and Supplementary Figure 12**).

Supplementary Figure 11. Colocalization of fPlas-gold with GFP-fused early endosomes as revealed with DFM and FM imaging. (a) DFM image of fPlas-gold; FM image of (b) fPlas-gold and (c) early endosomes; (d) overlay of a and b; (e) overlay of a and c; (e) overlay of b and c. To avoid influence of green GFP fluorescence on DFM signal of green spots, the green fluorescence was converted to blue color in e. DFM images of fPlas-gold were taken on the correlative microscope. FM images of fPlas-gold and GFP-fused early endosomes were taken on the confocal microscope layer by layer and then reconstructed.

Supplementary Figure 12. Colocalization of fPlas-gold with GFP-fused late endosomes as revealed with DFM and FM imaging. (a) DFM image of fPlas-gold; FM image of (b) fPlas-gold and (c) late endosomes; (d) overlay of a and b; (e) overlay of a and c; (e) overlay of b and c. To

avoid influence of green GFP fluorescence on DFM signal of green spots, the green fluorescence was converted to blue color in e. DFM images of fPlas-gold were taken on the correlative microscope. FM images of fPlas-gold and GFP-fused late endosomes were taken on the confocal microscope layer by layer and then reconstructed.

Responses to Reviewer #2:

Concern 1:

In the current work, the authors investigate internalization and intracellular traffic of a DNA-mediated dually emissive gold probe in living cells. The use of both plasmonic and fluorescent properties of the particles for monitoring intra-cellular processes is certainly a powerful method, which opens new advances in bio-imaging. However, one of the key prerequisites of correct interpretation of the results is monodispersity of the probe. However, supplementary figure 1(b) clearly shows that the gold particles differ in size. Moreover, even by inspecting the provided low resolution image one can see that size of some of the particles differ by a factor of 2. This raises the question on interpretation of the spectra shown in supplementary figure 3 (b) and (c). The clear deviation of the measured data from the theoretical model is most likely caused by the broad size distribution of the particles. This makes impossible to make any reliable conclusions from the data. How do the authors distinguish clustering of the particles from the variation of their size?

Response:

We highly appreciate the comments and suggestions from the Reviewer 2. We have performed additional experiments and improved our data.

The size deviation of AuNPs would not cause change of optical properties under DFM imaging. Additional data has been included in the revised manuscript, revealing that over 95% of AuNPs represents as green spots, based on the analysis of over 1,000 randomly selected particles (**Supplementary Figure 1c and 1d**).

Supplementary Figure 1. Characterization of f-Plas gold. (a) Nanoparticle absorbance spectra before (black) and after (red) DNA modification. (b) A representative TEM image and (c) three representative DFM images of fPlas-gold. (d) Over 1,000 dots in DFM images were analyzed, revealing that more than 95% of fPlas-gold in water on glass existed as green dots in DFM images.

Concern 2:

In my view, the manuscript contains excess of acronyms, which are so not commonly used in the fluorescence microscopy community. For instance, SP, LC, MSD, SC, ... This makes the manuscript harder to read. The authors could especially avoid overload of acronyms in the figure captions.

Response: We apologize for using excess of acronyms and making the manuscript hard to read. We have reduced the use of acronyms in the revised manuscript.

Concern 3.

Figure 1(d): why do the authors use a 3d representation of the plot? The plot can be shown exactly in the same way as the one in figure (e), which would clearly improve its readability.

Response:

We have replaced Figure 1d as the reviewer suggested.

Responses to Reviewer #3:

The manuscript by Lal & Fan et al. applied gold nanoparticles, which have the distinct ability to change spectra under different clustering status, to investigate nanoparticle clustering in the endocytosis process of living cells; also, by combining dark field microscopy, fluorescent imaging and electron microscopy, this manuscript carried out an in-depth study of the endocytosis-based transport of DNA-decorated gold nanoparticles. This Reviewer believes that this work is both novel and important. The novelty and importance of this work mainly come from: 1) although the phenomenon of spectrum change of gold nanoparticles upon aggregation is well-known and it has been used to study nanoparticle clustering before, its application for studying nanoparticle clustering in living cells is new, and this effort is of significant value considering the fact that nanoparticle clustering in living cells is highly important for nanobiotechnology and nanomedicine and yet is lack of careful studies. 2) Some of the findings are of great interests: after gold nanoparticles are internalized into cells, inside early endosomes are mainly single particles, and these particles are transported along microtubules to the perinuclear region mainly in single particle status; nanoparticle clustering occurs mainly in later stages. The experiments are well-designed and the presentation is clear. Thus, this Reviewer recommends Acceptance of this manuscript provided that the following specific points are properly addressed:

Concern 1:

In Figure 1, description of (f) appears to be missing in the figure legend.

Response:

Thank you for this constructive suggestion. Actually, the description of (f) follows the one of (b).

(b) DFM and (f) TEM images for time-evolution of fPlas-gold incubated with Hela cells.

Concern2:

In Figure 2, "microtubulin" should be "microtubule".

Response:

Thanks for this suggestion. We have modified it in the manuscript as suggested.

Concern3:

In the method to study colocalization, how are tMr values determined?

Response:

The tMr values were obtained using imageJ software. The values were exactly the thresholded Mander's coefficients. We have made this point clearer in the revised manuscript (**P7**) and cited a new reference (**Ref 42**):

“Time-lapse studies revealed the degree of co-localization of fPlas-gold with these cellular compartments, which were quantified by the thresholded Mander's colocalization coefficients (tMr)⁴² representing the percentage of the Cy3 fluorescence

overlaid with the GFP.”

Concern 4:

The proposed explanation on why the transport speed for gold nanoparticles as the cargo compared with that for a "normal" cargo was that the density of gold is higher than that of a "normal" cargo. This explanation is not convincing. In the cargo transport in living cells, gravity is not a main resistance to overcome. Rather, the main resistance comes from drag force (hydrodynamic friction due to viscosity), which is mainly related to the size and shape of the cargo, and is independent of the density. Take the 2-dimensional movement of cargo imaged and tracked in this manuscript as an example, the direction of gravity is in the direction perpendicular to the direction of cargo movement, and thus shouldn't cause significant resistance.

Response:

We appreciate this suggestion very much. We do agree with the reviewer on the interpretation of the transport speed, and accordingly modified it .

We further analyzed our TEM data and found out the vesicle size increased along with the numbers of encapsulated AuNPs (**Figure S21**). Only single particles were found in small vesicles while small and large clusters were observed in bigger vesicles.

Supplementary Figure 21. Dependence of vesicle sizes on the number of encapsulated AuNPs. TEM images of randomly selected (a) ten vesicles containing single particles and (b) ten vesicles containing clusters. (c) The measured vesicle diameters of single particles and clusters. Data were presented as the mean \pm SEM. ** $P < 0.01$, according to student-t test.

Furthermore, we examined the dynamics of fPlas-gold transport and movement of early endosomes, late endosomes and lysosomes (**Supplementary Figure 14 and Supplementary Video 2a-c**) and found that most fPlas-gold captured in early endosomes showed high mobility and most of the ones in late endosomes and lysosomes showed low mobility.

Supplementary Figure 14. Representative trajectories of intracellular fPlas-gold of high mobility (magenta) and low mobility (blue) in FM images. See Supplementary Video 2: $\Delta t = 3$ s, total time = 90 s for a (early endosomes) and b (late endosomes), 450 s for c (lysosomes).

Taken together, we conclude that the moving speed of vesicles is highly dependent on the vesicle size, as the reviewer suggested. We have included these data and provided discussion as shown below in the revised manuscript (**P8 & P11**).

P8: “We then examined the transport dynamics of fPlas-gold and movement of these three organelles (Figure S14 and Supplementary Video 2: $\Delta t = 3$ s, total time = 90 s for a and b, 450s for c), which consistently showed that most particles captured in early endosomes were of high mobility whereas those in late endosomes and lysosomes were of low mobility. We further note that the highest moving speed of fPlas-gold was of $\sim 0.6 \mu\text{m/s}$, several folds slower than typical transport speed of cargos (up to several $\mu\text{m/s}$), which might be associated with the sorting process of these inorganic particles.”

P11: “Notably, TEM analysis revealed that the vesicle size for single particles were significantly smaller than that for small and large clusters (Figure S21), suggesting that the number of fPlas-gold in these three organelles corresponds well with the vesicle size.”

Concern 5:

In Line 141, "nuclear region" should be "perinuclear region", as the particles didn't really enter the cell nucleus.

Response:

Thanks for this suggestion. We have modified it as suggested.

Concern 6:

In Line 25, "know" should be "known".

Response:

Thanks for this suggestion. We have modified it as suggested.

Concern 7:

In Line 257, "vascular" should be "vesicle".

Response:

Thanks for this suggestion. We have modified it as suggested.

Responses to Reviewer #4:

The paper describes an application of fluorescently labeled dsDNA-coated gold nanoparticles in studies of nanoparticle cell entry pathways and intracellular trafficking. The most interesting results are associated with observations of different motion patterns in cells using dark-field microscopy; this approach can potentially allow a quantitative statistical analysis of interactions between different endosomal vesicles. However, the authors have stopped short of carrying out such quantitative studies limiting their experiments to a set of observations/examples. Therefore, the paper is an interesting extension/continuation of work that has been previously done by other groups (representative studies are referenced by the authors). However, it does not have sufficient novelty to warrant publication in Nature Communications and it would be better suited for a specialty journal in the field of nanotechnology or imaging. Furthermore, there are multiple technical concerns that need to be addressed which are summarized below.

Concern 1:

Major Concerns deal with the validation of nanoparticles association with various endosomal compartments in cells:

All "spots" in dark-field microscopy (DFM) are categorized as single particles (SP), small clusters (SC) and large clusters (LC) solely based on their apparent color - green, yellow and red, respectively. How were these colors identified? In other words, how it was determined that a color is green, yellow or red. This is an important definition that has to be done in a quantitative way because many conclusions in the paper are based on it.

Response:

We thank reviewer 4 for this important question. We added a section in the Supplementary Information to address this question (**P8 in Supplementary Information, and associated supplementary figures**).

The green, yellow and bright yellow colors of fPlas-gold in DFM images are the colors that we visualized under the microscope, which are further corroborated by collecting their scattering spectra (see Supplementary Figure 4b). We then compared our naked-eye classification of randomly-selected 20 fPlas-gold clusters with the results based on scattering spectra, and found a good agreement (see Supplementary Figure 4d).

We also performed finite-difference time-domain (FDTD) simulation and found that the simulated scattering spectra were generally consistent with the measured ones (Supplementary Figure 4c).

Correlative imaging with SEM and DFM established the dependence of the color change on the aggregation states of fPlas-gold. A single particle of fPlas-gold exhibited green color in DFM images, whereas the color of clustered fPlas-gold gradually turned to yellow along with the increased number (n) of particles (from n=1 to n=10) (Supplementary Figure 4a). Yellow spots in DFM images are small clusters containing 2-5 single particles under SEM imaging and bright yellow spots are large

clusters containing more than 5 particles.

Finally, we performed a quantitative analysis on the correlation between the color of fPlas-gold in DFM images with their aggregation states. A wide-field DFM image containing approximately 100 particles was recorded (Supplementary Figure 3, top). This image was subsequently used as a pattern recognition template during the SEM analysis to locate and correlate particles of different aggregation states (Supplementary Figure 3, bottom). This study confirmed the robustness of color classification.

Supplementary Figure 3. Determination of fPlas-gold aggregation states. Top: a wide-field DFM image containing approximately 100 spots of different colors, Bottom: the ex-situ SEM image of the fPlas-gold particles recorded in the DFM image. The bright yellow spot in the red circle contained 8 fPlas-gold single particles.

Supplementary Figure 4. Optical property of fPlas-gold. (a) Representative SEM images of clusters of different sizes and the corresponding DFM images. (b) Scattering spectra detected by experiment and (c) simulated by FDTD for fPlas-gold shown in (a). The scattering spectra gradually shifted to the red with the increase of cluster size. (d) We compared naked-eye classification of randomly-selected 20 fPlas-gold clusters with the results based on scattering spectra, and found a good agreement.

Concern 2:

The authors have not fully explored the double contrast capability of their nanoparticles - fluorescent and DMF imaging have been carried out separately. The problem is that the authors have never established that their SP, SC and LC particles are actually associated with early endosomes, late endosomes and lysosomes, respectively, as it is claimed in data analysis/results. This correlation was assumed based on fluorescent co-localization measurements that have been carried out separately from DMF. Furthermore, there is a problem in interpretation of the co-localization studies in Figure 2. It appears from the images in Fig. 2c that nanoparticles (red fluorescence) are not co-localized with various endosomal compartments (green fluorescence) as it is claimed because there is no evident yellow color in the overlay images that would be an indication of a co-localization.

Response:

We thank this reviewer for these constructive comments. We have improved the quality of Figure 2c. We also performed additional experiments to further confirm our

conclusions. Importantly, we have demonstrated the correlation of fluorescence and DFM signals in this revision as shown below.

1) We observed Cy3 fluorescence and DFM signal of 50 nm AuNPs in HeLa cells within 24 h and did not find obvious signal separation, which strongly suggested the correlation of fluorescence signal and DFM signal. These data have been included in the revised manuscript (**P5, Figure S7**).

Supplementary Figure 7. Time-dependent cellular uptake of fPlas-gold. Representative examples of DFM images of HeLa cells incubated with 0.1 nM fPlas-gold for different time. Both DFM images and FM images were taken by using the correlative microscope.

2) We overlaid DFM and FM signals of fPlas-gold with FM signals of endosomes in fixed HeLa cells and found nice colocalization of them, as shown in the figures below. These data have been included in the revised manuscript (**Figure S11 and 12**).

Supplementary Figure 11. Colocalization of fPlas-gold with GFP-fused early endosomes as revealed with DFM and FM imaging. (a) DFM image of fPlas-gold; FM image of (b) fPlas-gold and (c) early endosomes; (d) overlay of a and b; (e) overlay of a and c; (e) overlay of b and c. To avoid influence of green GFP fluorescence on DFM signal of green spots, the green fluorescence was converted to blue color in e. DFM images of fPlas-gold were taken on the correlative microscope. FM images of fPlas-gold and GFP-fused early endosomes were taken on the confocal microscope layer by layer and then reconstructed.

Supplementary Figure 12. Colocalization of fPlas-gold with GFP-fused late endosomes as revealed with DFM and FM imaging. (a) DFM image of fPlas-gold; FM image of (b) fPlas-gold and (c) late endosomes; (d) overlay of a and b; (e) overlay of a and c; (e) overlay of b and c. To

avoid influence of green GFP fluorescence on DFM signal of green spots, the green fluorescence was converted to blue color in e. DFM images of fPlas-gold were taken on the correlative microscope. FM images of fPlas-gold and GFP-fused late endosomes were taken on the confocal microscope layer by layer and then reconstructed.

3) The movement of fPlas-gold and early endosomes, late endosomes and lysosomes in living cells were investigated, respectively, by detecting Cy3 fluorescence of fPlas-gold and fluorescence of these stained intracellular organelles. The observed results demonstrated that fPlas-gold and organelles move together, which strongly suggested capture of fPlas-gold by endosomes and lysosomes. These results were included in **Supplementary Videos 2a-c and Supplementary Figure 14**.

Supplementary Figure 14. Representative trajectories of intracellular fPlas-gold of high mobility (magenta) and low mobility (blue) in FM images. See Supplementary Video 2: $\Delta t = 3$ s, total time = 90 s for a (early endosomes) and b (late endosomes), 450 s for c (lysosomes).

In summary, we observed single particles, small clusters and large clusters captured in early endosomes, late endosomes and lysosomes, respectively, using TEM. We also observed colocalization of Cy3 fluorescence with GFP fluorescence of endosomes and lysotracker signal of lysosomes in fixed cells, and movement of Cy3 fluorescence with fluorescence of endosomes and lysosomes in living cells. Using DFM, we observed movement of single fPlas-gold and found the difference in the movement of single particles, small clusters and large clusters. Taken together, we provide new knowledge of nanoparticle entry, clustering and transport by using this new method.

Concern 3:

Fig. 3: similarly to the concern above, it appears that the correlation of stage I - low motility particles, stage 2 - high motility particles, and stage 3 - low motility particles with those bound to the cellular membrane, in early endosomes, and in lysosomes, respectively, is also assumed because there is no co-labeling of the endosomal compartments in these experiments.

Response:

We thank the reviewer for this excellent suggestion. We performed additional experiments to investigate the movement of fPlas-gold and early endosomes, late endosomes and lysosomes in living cells, respectively, by detecting Cy3 fluorescence

of fPlas-gold and fluorescence of these stained intracellular organelles. As shown in **Supplementary Videos 2 a-c and Supplementary Figures 14**, most of the fPlas-gold captured in early endosomes showed high mobility, and most of the ones in late endosomes and lysosomes showed low mobility, which is consistent with the data shown in Figure 3. We have included this discussion in the revised manuscript (**P8**).

P8: “We then examined the transport dynamics of fPlas-gold and movement of these three organelles (Figure S14 and Supplementary Video 2: $\Delta t = 3$ s, total time = 90 s for a and b, 450s for c), which consistently showed that most particles captured in early endosomes were of high mobility whereas those in late endosomes and lysosomes were of low mobility.”

Supplementary Figure 14. Representative trajectories of intracellular fPlas-gold of high mobility (magenta) and low mobility (blue) in FM images. See Supplementary Video 2: $\Delta t = 3$ s, total time = 90 s for a (early endosomes) and b (late endosomes), 450 s for c (lysosomes).

Minor comments:

Concern 4:

Suppl Fig 1: an increase in the absorbance of particle conjugates above 600nm can be indicative of some aggregation. TEM data cannot be used to evaluate particle aggregation state due to well-known sample preparation artifacts. The particles should be characterized by dynamic light scattering (DLS). In addition, Z-potential should be also determined because particle charge is critically important in interactions with cells.

Response:

We have included data of DLS and Z-potential in the revised manuscript (**P4**).

P4: “Compared to unmodified AuNPs, Z-potential of assembled fPlas-gold increased from -44.0 ± 0.4 mV to -34.4 ± 0.4 mV, and the value of dynamic light scattering (DLS) increased from 46.3 ± 0.5 nm to 71.6 ± 1.8 nm. The increase in the hydrodynamic diameter of AuNPs from DLS corresponds to an adlayer of DNA. We also note that fPlas-gold stayed in the near monodispersed state with small size deviation.”

Concern 5:

The statement (lines 126-127) - "Typically, we could easily distinguish SP (n=1; green), small clusters (SC; n=2-5; yellow) and large clusters (LC; n>5; bright yellow)

under DFM" - is not generally true because the color is also a function of changes in microenvironment surrounding gold nanoparticles that also needs to be taken into account.

Response:

Based on our theoretical calculation, the cell microenvironment should not influence the optical properties of fPlas-gold. This is further confirmed by our experimental observation that extra- and intra-cellular fPlas-gold exhibit same optical scattering properties. We picked scattering spectrum of the extracellular green dot in Supplementary Figure 4a and the intracellular green dot in Supplementary Figure 5a, and overlaid them together. As shown in those two figures below, the scattering spectra of the extracellular single particle overlaid with the one of intracellular single particle perfectly. Therefore, the cellular microenvironment did not affect the optical scattering properties (i.e. color) of fPlas-gold. These data have been included in the revised manuscript (**P5, and Figure S6**).

Supplementary Figure 6. Comparison of light scattering spectra of extracellular and intracellular single particle fPlas-gold. The scattering spectra of the green dot in Supplementary Figure 4a (extracellular, blue) and Supplementary Figure 5a (intracellular, red) matched well. The right figure was the zoomed-in spectra of left figure.

Concern 6:

tMr data analysis should be described.

Response:

We thank reviewer 4 for this suggestion. The tMr values we obtained using imageJ software are exactly the thresholded Mander's coefficients. We have made this point clearer in the revised manuscript (**P7**) and cited a new reference (**Ref 42**).

P7: "Time-lapse studies revealed the degree of co-localization of fPlas-gold with these cellular compartments, which were quantified by the thresholded Mander's colocalization coefficients (tMr)⁴² representing the percentage of the Cy3 fluorescence overlaid with the GFP."

Concern 7:

In Fig. 2d, it appears that only a fraction of nanoparticles is located in endosomes at any point of time. Where are the rest of the nanoparticles?

Response:

We have improved the quality of Figure 2d, which should give a better demonstration of colocalization of fPlas-gold with endosomes and lysosomes. Moreover, multi-pathways were involved in the cellular uptake process of fPlas-gold and thus some of the NPs would be located in other organelles which were not stained.

Concern 8:

Fig. 5: is there statistically significant difference between data for SP, SC and LC?

Response:

Thanks for this suggestion. We have added P value calculated by student-t test in Figure 5.

Reviewer #1 (Remarks to the Author):

The authors have added new experiments and performed exhaustive analysis to satisfactorily address all of the concerns brought up by the referees in the first round of review. However, even given these developments, I do not believe that the work outlined in the paper is novel enough to warrant publication in Nature Communications. Publication of this work in another journal would be more suitable.

Reviewer #2 (Remarks to the Author):

The authors have satisfactorily amended the manuscript. I recommend the manuscript for publication in its current form.

Reviewer #3 (Remarks to the Author):

The authors have adequately addressed this Reviewer's concerns, and I thus recommend acceptance of the revised manuscript.

Reviewer #4 (Remarks to the Author):

The authors have adequately addressed most of the technical comments. However, there are still some concerns regarding data and their interpretation:

- 1) The images showing co-registration of fluorescence and DFM images of fPlas-gold in Supplemental Figures 7, 11 and 12 do not look very convincing because they are acquired using 2D imaging that might not be sensitive enough to show separation of fluorescence and scattering signals from gold nanoparticles. High resolution confocal 3D sectioning would provide more convincing results.
- 2) Fluorescence and DF imaging of fPlas-gold movement has been conducted separately instead of correlative microscopy. So, these results are compared indirectly.
- 3) One of the conclusions that the motility of the particles is dependent on their clustering states has not been fully justified. Indeed, clustering occurs as nanoparticles move to late endosomes and lysosomes. So, the movement pattern could be also associated with differences in movement of different vesicles inside the cell and not just the clustering state of the particles. This possibility has not been discussed.

Although the study is overall well executed and provides interesting details about intracellular trafficking of DNA-coated gold nanoparticles, most of the main conclusions of this study do not appear to be novel. Indeed, dual DF and fluorescence imaging of gold nanoparticles with fluorescently labeled coatings has been carried out previously. Also, it has been demonstrated by a number of groups that gold nanoparticles can utilize multiple cell entry pathways followed by trafficking from early to late endosomes and to lysosomes. Furthermore, it was shown that this intracellular trafficking is associated with progressive particle clustering. Therefore, this reviewer believes that this study is better suited for a more specialized journal.

Reviewers' comments:

Reviewer 1 (Remarks to the Author):

Concern: *The authors have added new experiments and performed exhaustive analysis to satisfactory address all of the concerns brought up by the referees in the first round of review. However, even given these developments, I do not believe that the work outlined in the paper is novel enough to warrant publication in Nature Communications. Publication of this work in another journal would be more suitable.*

Response: We thank the reviewer for acknowledging our efforts to address all previous concerns. Our study reports unique relationship between clustering states of nanoparticles and their intracellular transport, the information mostly unavailable in the literature. Although, it would be difficult to defend “novelty” absolutely, we strongly believe that our work is novel enough to be published in *Nature Communications*. The rationale for our novelty compared to the previous studies is described below.

Briefly, in this work, we *for the first time established direct relationship between clustering states of nanoparticles and their intracellular transport* by using a new correlative microscopic imaging approach. Our technique allows us to monitor real-time clustering and intracellular movement of nanoparticles at the single-cell level. Please see the new section added in manuscript (P9-P10, highlighted in yellow, Figure 5a-c) and the Supplementary Information (Supplementary figures 18-20).

Studies on nanoparticle entry in live cells have become the recent focus since such knowledge would provide the mechanism for nanoparticle/virus entry, and designing novel nanotherapeutics. Previous studies in the literature generally fall into three categories based on imaging techniques employed, i.e. **fluorescence imaging**, **plasmonic imaging** and **correlative imaging**. However, none of these studies reports the whole process of how nanoparticles are transported into cells and clustered inside cells, the information obtained in our present work, as described below:

1. Fluorescence microscopy (FM) is widely used to investigate cellular endocytosis and movement of single nanoparticle/virus. However, information about the clustering states of intracellular particles has not been reported, most likely because of limited spatial resolution of the technique. Below shows a few representative work using FM.

Cell entry of single virus has been studied by using live-cell fluorescence imaging. For example, Zhuang et al.¹⁻³ reported a series of distinguished work on entry mechanisms of influenza viruses. They studied the transport, acidification, and fusion of single virus particles in living cells and dissected individual stages of the viral entry pathway by monitor receptor-mediated endocytosis of individual virus particles. Hurley et al.⁴ investigated host-pathogen interaction during viral entry with confocal microscopy, using giant unilamellar vesicles and host proteins of the endosomal sorting complex required for transport (ESCRT) Becker et al.⁵ observed transport of large viral nucleocapsids over long distances from the viral replication centers to the budding sites. *However, it is a challenge to probe the assembly of viruses at the single-particle level, because the background fluorescence from newly synthesized viral components makes it extremely difficult to monitor early assembly steps.*⁶

FM was also used to study entry and movement of nanoparticles. Dahan et al.⁷ tracked the motion of intracellular proteins, by characterizing the *in vivo* motion of individual Quantum Dots (QDs)-tagged kinesin motors in living cells. By using spinning disk confocal microscopy, Nie et al.⁸ followed the transport of peptide-conjugated QDs into live cells. Strano et al.⁹ tracked endocytosis and exocytosis of DNA wrapped single-walled carbon nanotubes in cells. Yang et al.¹⁰ introduced a three-dimensional multi-resolution method to observe the cellular binding and uptake of nanoparticles. We previously investigated cellular uptake and transport of DNA tetrahedron nanostructures.¹¹ Wright et al.¹² investigated size-dependent cellular uptake of DNA functionalized 10, 15, 20, 40, and 50 nm AuNPs using confocal microscopy combined with ICP-MS analysis.

However, with FM, information of clustering states of intracellular particles could not be obtained.

2. Plasmonic imaging is usually used to investigate nanoparticles aggregation in cells. El-Sayed et al.¹³⁻¹⁶ examined the accumulation of nanoparticles in cell or nucleus. Using multispectral plasmon coupling microscopy, Reinhard et al.¹⁷ examined the correlation between spectral response and cluster size and distinguished individual AgNPs from clusters of different association levels. Wei et al.¹⁸ observed three dimensional distribution of aggregated AuNPs in live cells using dark-field sectional optical microscopy and discussed the relationship between scattering colors and aggregated numbers of AuNPs.

However, as best as we know, there has not been existing work reporting real-time movement of plasmonic nanoparticles with different aggregation states.

3. Correlative imaging using DFM/EM or FM/EM has been used to examine endocytosis of nanoparticles. However, EM could only provide information on internalization, aggregation and localization of nanoparticles in fixed cells. As best as we know, **there has not been any detailed study on the effect of nanoparticle clustering on the intracellular transport.** We present several pertinent examples below.

Mirkin et al.¹⁹⁻²¹ studied cellular uptake, subcellular localization, intracellular transport and the endocytosis mechanism of DNA-modified nanoparticles, by using confocal microscopy and TEM. They found out endocytosis of those particles involved a lipid-raft-dependent, caveolae-mediated pathway. **However, they did, not address any clustering of particles in the cells.** Chan et al.^{22, 23} investigated the effects of size, shape and components of nanoparticles on their cellular uptake by using FM and TEM. They also observed accumulation of fluorescently labeled nanoparticles in liver using whole animal imaging although clustering of nanoparticles was not examined. Sönnichsen et al.²⁴ used optical dark-field microscopy in conjunction with TEM to investigate the uptake of AuNPs into epithelial cells with respect to shape, stabilizing agent, and surface charge. **However, they have not examined the intracellular transport of the nanoparticles,** they only quantified the number of nanoparticles present in cells and their degree of aggregation. Reinhard et al.²⁵ investigated the scavenger receptor mediated uptake and subsequent intracellular spatial distribution and clustering of AgNPs, by using the combination of DFM, FM and SEM. Ren et al.²⁶ reported a dark-field illumination-based scattering correlation spectroscopy (DFSCS) and developed a theoretical model for translational diffusion of nanoparticles. **However, they**

only observed cellular distribution of AuNPs with TEM and DFSCS, and calculated the diffusion efficient of AuNPs to conclude the intracellular environments complex and heterogeneous.

Compared to these existing studies, our manuscript for the first time established direct correlation between clustering states of nanoparticles and their intracellular transport at real-time, single-cell and single-particle level. Our findings provide important and valuable information on endocytosis of nanoparticles.

We would like to emphasize that such new information on the intracellular trafficking of endocytosed nanoparticles would not be revealed without using the correlative microscope developed in this work. **Hence, our work not only represents a technical advance, it also describes new information on nanoparticle entry and transport.** We believe such studies would set a new paradigm for designing high-efficiency targeted delivery carriers and nanomedicine. We have addressed these concerns by adding a section in discussion (P12-P13, highlighted in yellow).

1. Lakadamyali, M., M.J. Rust, and X. Zhuang, *Ligands for clathrin-mediated endocytosis are differentially sorted into distinct populations of early endosomes*. Cell, 2006. **124** (5): p. 997-1009.
2. Lakadamyali, M. and X. Zhuang, *Visualizing infection of individual influenza viruses*. Proceedings of the National Academy of Sciences of the United States of America, 2003. **100** (16): p. 9280-5.
3. Rust, M.J., et al., *Assembly of endocytic machinery around individual influenza viruses during viral entry*. Nature Structural & Molecular Biology, 2004. **11** (6): p. 567-73.
4. Carlson, L.A. and J.H. Hurley, *In vitro reconstitution of the ordered assembly of the endosomal sorting complex required for transport at membrane-bound HIV-1 Gag clusters*. Proceedings of the National Academy of Sciences of the United States of America, 2012. **109** (42): p. 16928-33.
5. Schudt, G., et al., *Live-cell imaging of Marburg virus-infected cells uncovers actin-dependent transport of nucleocapsids over long distances*. Proceedings of the National Academy of Sciences of the United States of America, 2013. **110** (35): p. 14402-7.
6. Brandenburg, B. and X. Zhuang, *Virus trafficking - learning from single-virus tracking*. Nature Reviews Microbiology, 2007. **5** (3): p. 197-208.
7. Courty, S., et al., *Tracking individual proteins in living cells using single quantum dot imaging*. Nano Letters, 2006. **6** (7): p. 1491-5.
8. Ruan, G., et al., *Imaging and tracking of tat peptide-conjugated quantum dots in living cells: new insights into nanoparticle uptake, intracellular transport, and vesicle shedding*. Journal of the American Chemical Society, 2007. **129** (129): p. 14759-66.
9. Jin, H., D.A. Heller, and M.S. Strano, *Single-Particle Tracking of Endocytosis and Exocytosis of Single-Walled Carbon Nanotubes in NIH-3T3 Cells*. 2008. **8** (6): p. 1577-85.
10. Welsher, K. and H. Yang, *Multi-resolution 3D visualization of the early stages of cellular uptake of peptide-coated nanoparticles*. Nature Nanotechnology, 2014. **9** (3): p. 198-203.
11. Liang, L., et al., *Single-particle tracking and modulation of cell entry pathways of a tetrahedral*

- DNA nanostructure in live cells*. *Angewandte Chemie*, 2014. **53** (30): p. 7745-50.
12. Wong, A.C. and D.W. Wright, *Size-Dependent Cellular Uptake of DNA Functionalized Gold Nanoparticles*. *Small*, 2016. **12** (40): p. 5592-5600.
 13. Ali, M.R.K., S.R. Panikkanvalappil, and M.A. Elsayed, *Enhancing the Efficiency of Gold Nanoparticles Treatment of Cancer by Increasing Their Rate of Endocytosis and Cell Accumulation Using Rifampicin*. *Journal of the American Chemical Society*, 2014. **136** (12): p. 4464-7.
 14. Austin, L.A., et al., *Plasmonic Imaging of Human Oral Cancer Cell Communities During Programmed Cell Death by Nuclear Targeting Silver Nanoparticles*. *Journal of the American Chemical Society*, 2011. **133** (44): p. 17594-7.
 15. El-Sayed, I.H., X. Huang, and M.A. El-Sayed, *Surface plasmon resonance scattering and absorption of anti-EGFR antibody conjugated gold nanoparticles in cancer diagnostics: applications in oral cancer*. *Nano Letters*, 2005. **5** (5): p. 829-34.
 16. Kang, B., M.A. Mackey, and M.A. Elsayed, *Nuclear Targeting of Gold Nanoparticles in Cancer Cells Induces DNA Damage, Causing Cytokinesis Arrest and Apoptosis*. *Journal of the American Chemical Society*, 2010. **132** (5): p. 1517-9.
 17. Wang, H., et al., *Optical Sizing of Immunolabel Clusters through Multispectral Plasmon Coupling Microscopy*. *Nano Letters*, 2016. **11** (2): p. 498-504.
 18. Wang, S.H., et al., *Chromatogram Analysis on Revealing Aggregated Number and Location of Gold Nanoparticles Within Living Cells*. *Plasmonics*, 2015. **10** (4): p. 873-80.
 19. Wu, X.A., et al., *Intracellular fate of spherical nucleic acid nanoparticle conjugates*. *Journal of the American Chemical Society*, 2014. **136** (21): p. 7726-33.
 20. Cutler, J.I., et al., *Polyvalent Nucleic Acid Nanostructures*. *Journal of the American Chemical Society*, 2011. **133** (24): p. 9254-7.
 21. Choi, C.H., et al., *Mechanism for the endocytosis of spherical nucleic acid nanoparticle conjugates*. *Proceedings of the National Academy of Sciences of the United States of America*, 2013. **110** (19): p. 7625-30.
 22. Ohta, S., D. Glancy, and W.C. Chan, *DNA-controlled dynamic colloidal nanoparticle systems for mediating cellular interaction*. *Science*, 2016. **351** (6275): p. 841-5.
 23. Chou, L.Y.T., K. Zagorovsky, and W.C.W. Chan, *DNA assembly of nanoparticle superstructures for controlled biological delivery and elimination*. *Nature Nanotechnology*, 2014. **9** (2): p. 148-55.
 24. Rosman, C., et al., *A new approach to assess gold nanoparticle uptake by mammalian cells: combining optical dark-field and transmission electron microscopy*. *Small*, 2012. **8** (23): p. 3683-90.
 25. Wang, H., L. Wu, and B.M. Reinhard, *Scavenger Receptor Mediated Endocytosis of Silver Nanoparticles into J774A.1 Macrophages Is Heterogeneous*. *ACS Nano*, 2012. **6** (8): p. 7122-32.
 26. Liu, H., C. Dong, and J. Ren, *Tempo-spatially resolved scattering correlation spectroscopy under dark-field illumination and its application to investigate dynamic behaviors of gold nanoparticles in live cells*. *Journal of the American Chemical Society*, 2014. **136** (7): p. 2775-85.

Reviewer 2 (Remarks to the Author):

Remarks: *The authors have satisfactorily amended the manuscript. I recommend the manuscript for publication in its current form.*

Response: We sincerely appreciate your encouragement and acceptance of the findings described in this work. Thank you!

Reviewer 3 (Remarks to the Author):

Remarks: *The authors have adequately addressed this Reviewer's concerns, and I thus recommend acceptance of the revised manuscript.*

Response: We sincerely appreciate your encouragement and acceptance of the findings described in this work. Thank you!

Reviewer 4 (Remarks to the Author):

The authors have adequately addressed most of the technical comments. However, there are still some concerns regarding data and their interpretation:

Concern 1) *The images showing co-registration of fluorescence and DFM images of fPlas-gold in Supplemental Figures 7, 11 and 12 do not look very convincing because they are acquired using 2D imaging that might not be sensitive enough to show separation of fluorescence and scattering signals from gold nanoparticles. High resolution confocal 3D sectioning would provide more convincing results.*

Response: We thank the reviewer for this constructive suggestion. To address this concern, we performed high resolution confocal 3D sectioning to obtain fluorescence signals of fPlas-gold and organelles and then reconstructed the 3D images to a 2D image and overlaid it with the DFM image (Supplemental Figures 11, 12). We added these new data in Supplemental Figures 11g, 11h, 12g, 12h as the reviewer suggested (highlighted in yellow).

Supplementary Figure 11.

Colocalization of fPlas-gold with GFP-fused early endosomes as revealed with DFM and FM imaging. (a) DFM image of fPlas-gold; FM image of (b) fPlas-gold and (c) early endosomes; (d) overlay of a and b; (e) overlay of a and c; (f) overlay of b and c. To avoid influence of green GFP fluorescence on DFM signal of green spots, the green fluorescence was converted to blue color in e. Confocal 3D sectioning images of GFP-fused early endosomes (g) and fPlas-gold (h). Note: DFM images of fPlas-gold (a) were taken on the correlative microscope. FM images of fPlas-gold (b) and GFP-fused early endosomes (c) were taken on the confocal microscope layer by layer (see images g and h) and then reconstructed.

Supplementary Figure 12.

Colocalization of fPlas-gold with GFP-fused late endosomes as revealed with DFM and FM imaging. (a) DFM image of fPlas-gold; FM image of (b) fPlas-gold and (c) early endosomes; (d) overlay of a and b; (e) overlay of a and c; (f) overlay of b and c. To avoid influence of green GFP fluorescence on DFM signal of green spots, the green fluorescence was converted to

blue color in e. **Confocal 3D sectioning images of GFP-fused late endosomes (g) and fPlas-gold (h).**
Note: DFM images of fPlas-gold (a) were taken on the correlative microscope. FM images of fPlas-gold (b) and GFP-fused late endosomes (c) were taken on the confocal microscope layer by layer (see images g and h) and then reconstructed.

Concern 2) Fluorescence and DF imaging of fPlas-gold movement has been conducted separately instead of correlative microscopy. So, these results are compared indirectly.

Response: The correlative FM/DFM microscope employed in this work does not support simultaneous imaging with these two modes (see optical setup shown in Supplementary Scheme 3).

However, our correlative imaging technique can switch between FM and DFM within 30 seconds and image the same area of interest. Therefore, our technique provides complementary information of the movement of fPlas-gold nanoparticle and their clustering states that reveals the mechanism of intracellular trafficking. For example, we used DFM imaging to monitor the movement of fPlas-gold of different clustering states; whereas FM imaging was employed to investigate the intracellular localization of fPlas-gold.

By monitoring single cells with both FM and DFM, we can obtain both dynamic and static information of the intracellular transportation of fPlas-gold particles, which would not be possible with a single-mode microscope. Hence, although the DFM and FM data were not obtained simultaneously, our dual-mode imaging technique opens new opportunities for studying cellular events at the single-cell level.

Supplementary Scheme 3. Schematic illustration of the optical pathway diagram of the correlative microscope employed in the present study.

Concern 3)

A) One of the conclusions that the motility of the particles is dependent on their clustering states has not been fully justified. Indeed, clustering occurs as nanoparticles move to late endosomes and lysosomes. So, the movement pattern could be also associated with differences in movement of different vesicles inside the cell and not just the clustering state of the particles. This possibility has not been discussed.

B) Although the study is overall well executed and provides interesting details about intracellular trafficking of DNA-coated gold nanoparticles, most of the main conclusions of this study do not appear to be novel. Indeed, dual DF and fluorescence imaging of gold nanoparticles with fluorescently labeled coatings has been carried out previously. Also, it has been demonstrated by a number of groups that gold nanoparticles can utilize multiple cell entry pathways followed by trafficking from early to late endosomes and to lysosomes. Furthermore, it was shown that this intracellular trafficking is associated with progressive particle clustering. Therefore, this reviewer believes that this study is better suited for a more specialized journal.

Response:

A) We thank the reviewer for raising this important concern. We have examined the relationship of movement between fPlas-gold of different clustering states and organelles (early endosomes, late endosomes and lysosomes). We analyzed 100 randomly selected particles were analyzed in each set of data for comparison. Importantly, the single-particle analysis revealed that, although the intracellular movement of fPlas-gold is directly associated with the clustering states (see new data shown in Figure 5a-c), it does not show direct correlation with the movement of organelles (Supplementary Figure 18-20). Therefore, while the motility of fPlas-gold might have some relationship with the movement of vesicles, it is primarily dependent on their clustering states.

We added a section in manuscript (P9-P10, highlighted in yellow, Figure 5a-c) and the Supplementary Information (Supplementary figures 18-20) to address these concerns.

B) Our study reports unique relationship between clustering states of nanoparticles and their intracellular transport, the information mostly unavailable in the literature. Although, it would be difficult to defend “novelty” absolutely, we strongly believe that our work is novel enough to be published in *Nature Communications*. The rationale for our novelty compared to the previous studies is described below.

In this work, we for the first time established direct relationship between clustering states of nanoparticles and their intracellular transport, by using a new correlative microscopic imaging approach. Our technique allows us to monitor real-time clustering and intracellular movement of nanoparticles at the single-cell level.

Studies on nanoparticle entry in live cells have become the recent focus since such knowledge would provide the mechanism for nanoparticle/virus entry, and designing novel nanotherapeutics. Previous studies in the literature generally fall into three categories based on imaging techniques employed, i.e. **fluorescence imaging**, **plasmonic imaging** and **correlative imaging**. However, none of these studies reports the whole process of how

nanoparticles are transported into cells and clustered inside cells, the information obtained in our present work, as described below:

1. Fluorescence microscopy (FM) is widely used to investigate cellular endocytosis and movement of single nanoparticle/virus. However, information about the clustering states of intracellular particles has not been reported, most likely because of limited spatial resolution of the technique. Below shows a few representative work using FM.

Cell entry of single virus has been studied by using live-cell fluorescence imaging. For example, Zhuang et al.¹⁻³ reported a series of distinguished work on entry mechanisms of influenza viruses. They studied the transport, acidification, and fusion of single virus particles in living cells and dissected individual stages of the viral entry pathway by monitor receptor-mediated endocytosis of individual virus particles. Hurley et al.⁴ investigated host-pathogen interaction during viral entry with confocal microscopy, using giant unilamellar vesicles and host proteins of the endosomal sorting complex required for transport (ESCRT) Becker et al.⁵ observed transport of large viral nucleocapsids over long distances from the viral replication centers to the budding sites. *However, it is a challenge to probe the assembly of viruses at the single-particle level, because the background fluorescence from newly synthesized viral components makes it extremely difficult to monitor early assembly steps.*⁶

FM was also used to study entry and movement of nanoparticles. Dahan et al.⁷ tracked the motion of intracellular proteins, by characterizing the *in vivo* motion of individual Quantum Dots (QDs)-tagged kinesin motors in living cells. By using spinning disk confocal microscopy, Nie et al.⁸ followed the transport of peptide-conjugated QDs into live cells. Strano et al.⁹ tracked endocytosis and exocytosis of DNA wrapped single-walled carbon nanotubes in cells. Yang et al.¹⁰ introduced a three-dimensional multi-resolution method to observe the cellular binding and uptake of nanoparticles. We previously investigated cellular uptake and transport of DNA tetrahedron nanostructures.¹¹ Wright et al.¹² investigated size-dependent cellular uptake of DNA functionalized 10, 15, 20, 40, and 50 nm AuNPs using confocal microscopy combined with ICP-MS analysis.

However, with FM, information of clustering states of intracellular particles could not be obtained.

2. Plasmonic imaging is usually used to investigate nanoparticles aggregation in cells. El-Sayed et al.¹³⁻¹⁶ examined the accumulation of nanoparticles in cell or nucleus. Using multispectral plasmon coupling microscopy, Reinhard et al.¹⁷ examined the correlation between spectral response and cluster size and distinguished individual AgNPs from clusters of different association levels. Wei et al.¹⁸ observed three dimensional distribution of aggregated AuNPs in live cells using dark-field sectional optical microscopy and discussed the relationship between scattering colors and aggregated numbers of AuNPs.

However, as best as we know, there has not been existing work reporting real-time movement of plasmonic nanoparticles with different aggregation states.

3. Correlative imaging using DFM/EM or FM/EM has been used to examine endocytosis of nanoparticles. However, EM could only provide information on internalization, aggregation and localization of nanoparticles in fixed cells. As best as we know, **there has not been any**

detailed study the effect of nanoparticle clustering on the intracellular transport. We present several pertinent examples below.

Mirkin et al.¹⁹⁻²¹ studied cellular uptake, subcellular localization, intracellular transport and the endocytosis mechanism of DNA-modified nanoparticles, by using confocal microscopy and TEM. They found out endocytosis of those particles involved a lipid-raft-dependent, caveolae-mediated pathway. **However, they did, not address any clustering of particles in the cells.** Chan et al.^{22, 23} investigated the effects of size, shape and components of nanoparticles on their cellular uptake by using FM and TEM. They also observed accumulation of fluorescently labeled nanoparticles in liver using whole animal imaging although clustering of nanoparticles was not examined. Sönnichsen et al.²⁴ used optical dark-field microscopy in conjunction with TEM to investigate the uptake of AuNPs into epithelial cells with respect to shape, stabilizing agent, and surface charge. **However they have not examined the intracellular transport of the nanoparticles,** they only quantified the number of nanoparticles present in cells and their degree of aggregation. Reinhard et al.²⁵ investigated the scavenger receptor mediated uptake and subsequent intracellular spatial distribution and clustering of AgNPs, by using the combination of DFM, FM and SEM. Ren et al.²⁶ reported a dark-field illumination-based scattering correlation spectroscopy (DFSCS) and developed a theoretical model for translational diffusion of nanoparticles. **However, they only observed cellular distribution of AuNPs with TEM and DFSCS, and calculated the diffusion efficient of AuNPs to conclude the intracellular environments complex and heterogeneous.**

Compared to these existing studies, **our manuscript for the first time established direct correlation between clustering states of nanoparticles and their intracellular transport at real-time, single-cell and single-particle level.** Our findings provide important and valuable information on endocytosis of nanoparticles.

We would like to emphasize that such new information on the intracellular trafficking of endocytosed nanoparticles would not be revealed without using the correlative microscope developed in this work. ***Hence, our work not only represents a technical advance, it also describes new information on nanoparticle entry and transport.*** We believe such studies would set a new paradigm for designing high-efficiency targeted delivery carriers and nanomedicine. We have addressed these concerns by adding a section in discussion (P12-P13, highlighted in yellow).

1. Lakadamyali, M., M.J. Rust, and X. Zhuang, *Ligands for clathrin-mediated endocytosis are differentially sorted into distinct populations of early endosomes.* Cell, 2006. **124** (5): p. 997-1009.
2. Lakadamyali, M. and X. Zhuang, *Visualizing infection of individual influenza viruses.* Proceedings of the National Academy of Sciences of the United States of America, 2003. **100** (16): p. 9280-5.
3. Rust, M.J., et al., *Assembly of endocytic machinery around individual influenza viruses during viral entry.* Nature Structural & Molecular Biology, 2004. **11** (6): p. 567-73.
4. Carlson, L.A. and J.H. Hurley, *In vitro reconstitution of the ordered assembly of the endosomal sorting complex required for transport at membrane-bound HIV-1 Gag clusters.* Proceedings of the National Academy of Sciences of the United States of America, 2012. **109** (42): p.

- 16928-33.
5. Schudt, G., et al., *Live-cell imaging of Marburg virus-infected cells uncovers actin-dependent transport of nucleocapsids over long distances*. Proceedings of the National Academy of Sciences of the United States of America, 2013. **110** (35): p. 14402-7.
 6. Brandenburg, B. and X. Zhuang, *Virus trafficking - learning from single-virus tracking*. Nature Reviews Microbiology, 2007. **5** (3): p. 197-208.
 7. Courty, S., et al., *Tracking individual proteins in living cells using single quantum dot imaging*. Nano Letters, 2006. **6** (7): p. 1491-5.
 8. Ruan, G., et al., *Imaging and tracking of tat peptide-conjugated quantum dots in living cells: new insights into nanoparticle uptake, intracellular transport, and vesicle shedding*. Journal of the American Chemical Society, 2007. **129** (129): p. 14759-66.
 9. Jin, H., D.A. Heller, and M.S. Strano, *Single-Particle Tracking of Endocytosis and Exocytosis of Single-Walled Carbon Nanotubes in NIH-3T3 Cells*. 2008. **8** (6): p. 1577-85.
 10. Welsher, K. and H. Yang, *Multi-resolution 3D visualization of the early stages of cellular uptake of peptide-coated nanoparticles*. Nature Nanotechnology, 2014. **9** (3): p. 198-203.
 11. Liang, L., et al., *Single-particle tracking and modulation of cell entry pathways of a tetrahedral DNA nanostructure in live cells*. Angewandte Chemie, 2014. **53** (30): p. 7745-50.
 12. Wong, A.C. and D.W. Wright, *Size-Dependent Cellular Uptake of DNA Functionalized Gold Nanoparticles*. Small, 2016. **12** (40): p. 5592-5600.
 13. Ali, M.R.K., S.R. Panikkanvalappil, and M.A. Elsayed, *Enhancing the Efficiency of Gold Nanoparticles Treatment of Cancer by Increasing Their Rate of Endocytosis and Cell Accumulation Using Rifampicin*. Journal of the American Chemical Society, 2014. **136** (12): p. 4464-7.
 14. Austin, L.A., et al., *Plasmonic Imaging of Human Oral Cancer Cell Communities During Programmed Cell Death by Nuclear Targeting Silver Nanoparticles*. Journal of the American Chemical Society, 2011. **133** (44): p. 17594-7.
 15. El-Sayed, I.H., X. Huang, and M.A. El-Sayed, *Surface plasmon resonance scattering and absorption of anti-EGFR antibody conjugated gold nanoparticles in cancer diagnostics: applications in oral cancer*. Nano Letters, 2005. **5** (5): p. 829-34.
 16. Kang, B., M.A. Mackey, and M.A. Elsayed, *Nuclear Targeting of Gold Nanoparticles in Cancer Cells Induces DNA Damage, Causing Cytokinesis Arrest and Apoptosis*. Journal of the American Chemical Society, 2010. **132** (5): p. 1517-9.
 17. Wang, H., et al., *Optical Sizing of Immunolabel Clusters through Multispectral Plasmon Coupling Microscopy*. Nano Letters, 2016. **11** (2): p. 498-504.
 18. Wang, S.H., et al., *Chromatogram Analysis on Revealing Aggregated Number and Location of Gold Nanoparticles Within Living Cells*. Plasmonics, 2015. **10** (4): p. 873-80.
 19. Wu, X.A., et al., *Intracellular fate of spherical nucleic acid nanoparticle conjugates*. Journal of the American Chemical Society, 2014. **136** (21): p. 7726-33.
 20. Cutler, J.I., et al., *Polyvalent Nucleic Acid Nanostructures*. Journal of the American Chemical Society, 2011. **133** (24): p. 9254-7.
 21. Choi, C.H., et al., *Mechanism for the endocytosis of spherical nucleic acid nanoparticle conjugates*. Proceedings of the National Academy of Sciences of the United States of America, 2013. **110** (19): p. 7625-30.
 22. Ohta, S., D. Glancy, and W.C. Chan, *DNA-controlled dynamic colloidal nanoparticle systems*

- for mediating cellular interaction*. Science, 2016. **351** (6275): p. 841-5.
23. Chou, L.Y.T., K. Zagorovsky, and W.C.W. Chan, *DNA assembly of nanoparticle superstructures for controlled biological delivery and elimination*. Nature Nanotechnology, 2014. **9** (2): p. 148-55.
 24. Rosman, C., et al., *A new approach to assess gold nanoparticle uptake by mammalian cells: combining optical dark-field and transmission electron microscopy*. Small, 2012. **8** (23): p. 3683–90.
 25. Wang, H., L. Wu, and B.M. Reinhard, *Scavenger Receptor Mediated Endocytosis of Silver Nanoparticles into J774A.1 Macrophages Is Heterogeneous*. Acs Nano, 2012. **6** (8): p. 7122-32.
 26. Liu, H., C. Dong, and J. Ren, *Tempo-spatially resolved scattering correlation spectroscopy under dark-field illumination and its application to investigate dynamic behaviors of gold nanoparticles in live cells*. Journal of the American Chemical Society, 2014. **136** (7): p. 2775-85.

Figure 5. Movement of fPlas-gold in cells. (a) Scatter plots showing passage length and maximum of the frame to frame instantaneous speed for each single particle (blue), small cluster (green) and large cluster (red). The rectangles represent the major distribution of these particles, containing 79 single particles (blue), 92 small clusters (green) and 75 large clusters (red), respectively. Left panel: histogram showing the distribution of maximum speed of single particles (blue), small clusters (green) and large clusters (red). Lower panel: histogram showing the distribution of passage length of single particles (blue), small clusters (green) and large clusters (red). Data were collected from 100 randomly selected spots in three independent experiments. (b) Average passage length and (c) average maximum speed of fPlas-gold (Data obtained from single-particle analysis shown in (a), and were presented as the mean \pm SEM). (d) Average speed (v) of fPlas-gold in cells. Particles observed using DFM were classified as single particles, small clusters and large clusters while the ones observed using FM were classified as high- and low-mobility ones, respectively. Corresponding curves of MSD are shown in Supplementary Figure 16. Data were collected from 50 spots in three independent experiments for each group and presented as the mean \pm SEM. * $P < 0.05$, ** $P < 0.01$, according to student-t test.

Supplementary Figure 18. Movement of single particles and early endosomes in cells. (a) Scatter plots showing passage length and Maximum of frame to frame instantaneous speed for each single particle (blue) and early endosome (red). Left panel: histogram showing the distribution of speed. Lower panel: histogram showing the distribution of passage length. (b) Average passage length (left) and average maximum speed (right) of single particles and early endosomes (Data obtain from single-particle analysis shown in (a), and were presented as the mean \pm SEM. * P <0.05, ** P <0.01, according to student-t test). Data were collected from 100 randomly selected spots in three independent experiments for each group.

Supplementary Figure 19. Movement of small clusters and late endosomes in cells. (a) Scatter plots showing passage length and Maximum of frame to frame instantaneous speed for each small cluster (blue) and late endosome (red). left panel: histogram showing the distribution of speed. Lower panel: histogram showing the distribution of passage length. (b) Average passage length (left) and average maximum speed (right) of small clusters and late endosomes (Data obtain from single-particle analysis shown in (a), and were presented as the mean \pm SEM. * $P < 0.05$, ** $P < 0.01$, according to student-t test). Data were collected from 100 randomly selected spots in three independent experiments for each group.

Supplementary Figure 20. Movement of large clusters and lysosomes in cells. (a) Scatter plots showing passage length and Maximum of frame to frame instantaneous speed for each large cluster (blue) and lysosome (red). Left panel: histogram showing the distribution of speed. Lower panel: histogram showing the distribution of passage length. (b) Average passage length (left) and average maximum speed (right) of large clusters and lysosomes (Data obtain from single-particle analysis shown in (a), and were presented as the mean \pm SEM. * P <0.05, ** P <0.01, according to student-t test). Data were collected from 100 randomly selected spots in three independent experiments for each group.

Reviewer #1 (Remarks to the Author):

The authors make a compelling argument for the novelty of their research. In addition, the revised manuscript includes further experiments that support their claims. I recommend publication of the revised manuscript.

Reviewer #4 (Remarks to the Author):

Additional data have strengthened this detailed and comprehensive study that offers some interesting details about specifics of nanoparticle trafficking in live cells. There is one conclusion that this reviewer does not agree with. The authors state on page 12 that: "Comparison of the movement between particles and organelles did not show direct correlation, indicating the organelle type does not significantly influence the particle movement." However, the data appear to suggest quite the opposite. The Supplemental figures 18-20 show that the average maximum speed and the average distance for both the particles and the organelles follow the same trend of an overall decrease in progression from early endosomes to lysosomes. Although, as pointed out by the authors movement parameters within the same organelle type can be statistically different between organelles carrying particles and organelles without particles. This point needs to be more clearly addressed/discussed before potential publication of this work.

Reviewers' comments:

Reviewer 1 (Remarks to the Author):

Concern: *The authors make a compelling argument for the novelty of their research. In addition, the revised manuscript includes further experiments that support their claims. I recommend publication of the revised manuscript.*

Response: We sincerely appreciate your encouragement and acceptance of the findings described in this work. Thank you!

Reviewer 4 (Remarks to the Author):

Concern: *Additional data have strengthened this detailed and comprehensive study that offers some interesting details about specifics of nanoparticle trafficking in live cells. There is one conclusion that this reviewer does not agree with. The authors state on page 12 that: "Comparison of the movement between particles and organelles did not show direct correlation, indicating the organelle type does not significantly influence the particle movement." However, the data appear to suggest quite the opposite. The Supplemental figures 18-20 show that the average maximum speed and the average distance for both the particles and the organelles follow the same trend of an overall decrease in progression from early endosomes to lysosomes. **Although, as pointed out by the authors movement parameters within the same organelle type can be statistically different between organelles carrying particles and organelles without particles.** This point needs to be more clearly addressed/discussed before potential publication of this work.*

Response: We thank the reviewer for acknowledging our effort to perform additional experiments and analysis to address all previous concerns. We also thank the reviewer for this constructive suggestion.

The motility of fPlas-gold might have some relationship with the movement of vesicles however, our reevaluation as suggested by the reviewer seems to suggest that the motility of fPlas-gold is primarily dependent on their clustering states. To support this notion, we compared the movement of 100 randomly selected early endosomes, late endosomes and lysosomes. We did not find significant difference between the movement of early and late endosomes (Supplemental Figure 21). On the other hand, the movement of single particles and small clusters showed a significant difference (Figure 5). These observations suggest the clustering state of fPlas-gold play a more important role than the organelle type in the intracellular movement of fPlas-gold. We added these new data in Supplementary Figure 21 and added a section in manuscript (P10, line 10-14; P12, line 11-13) to address this concern.

We sincerely appreciate encouragement of this reviewer, which significantly strengthened our manuscript. Thank you!

Supplementary Figure 21. Movement of early endosomes, late endosomes and lysosomes in cells. (a) Scatter plots showing passage length and Maximum of frame to frame instantaneous speed for each early endosome (blue), late endosome (green) and lysosome (red). Left panel: histogram showing the distribution of speed. Lower panel: histogram showing the distribution of passage length. (b) Average passage length and (c) average maximum speed of early endosomes, late endosomes and lysosomes (Data obtain from single-particle analysis shown in (a), and were presented as the mean \pm SEM. *P<0.05, **P<0.01, according to student-t test). Data were collected from 100 randomly selected spots in three independent experiments for each group.

Reviewer #4 (Remarks to the Author):

I would recommend the manuscript for publication in its current form.

Reviewers' comments:

Reviewer 4 (Remarks to the Author):

Concern: *I would recommend the manuscript for publication in its current form.*

Response: We sincerely appreciate your encouragement and acceptance of the findings described in this work. Thank you!